# Statistical Insights into HSIC in High Dimensions

**Tao Zhang**
School of Information Management and Engineering
Shanghai University of Finance and Economics
Shanghai 200433, China
`dalizhangt@163.com`

**Yaowu Zhang**[*]
School of Information Management and Engineering
MoE Key Laboratory of Interdisciplinary Research of Computation and Economics
Shanghai University of Finance and Economics
Shanghai 200433, China
`zhang.yaowu@mail.shufe.edu.cn`

**Tingyou Zhou**
School of Data Sciences
Zhejiang University of Finance and Economics
Hangzhou 310018, China
`zhoutingyou@zufe.edu.cn`

## Abstract

Measuring the nonlinear dependence between random vectors and testing for their statistical independence is a fundamental problem in statistics. One of the most popular dependence measures is the Hilbert-Schmidt independence criterion (HSIC), which has attracted increasing attention in recent years. However, most existing works have focused on either fixed or very high-dimensional covariates. In this work, we bridge the gap between these two scenarios and provide statistical insights into the performance of HSIC when the dimensions grow at different rates. We first show that, under the null hypothesis, the rescaled HSIC converges in distribution to a standard normal distribution. Then we provide a general condition for the HSIC based tests to have nontrivial power in high dimensions. By decomposing this condition, we illustrate how the ability of HSIC to measure nonlinear dependence changes with increasing dimensions. Moreover, we demonstrate that, depending on the sample size, the covariate dimensions and the dependence structures within covariates, the HSIC can capture different types of associations between random vectors. We also conduct extensive numerical studies to validate our theoretical results.

## 1 Introduction

Let $\mathbf{x} = (X_1, \ldots, X_p)^{\mathrm{T}} \in \mathbb{R}^p$ and $\mathbf{y} = (Y_1, \ldots, Y_q)^{\mathrm{T}} \in \mathbb{R}^q$ be two random vectors. The problems of measuring nonlinear dependence between $\mathbf{x}$ and $\mathbf{y}$ and testing for their independence are fundamental and have a wide range of applications in statistics and machine learning. For example, dependence measures can be applied in feature screening (Fan et al., 2020b), model checking (Clarke et al., 2018), graphical models (Maathuis et al., 2018), and causal inference (Imbens & Rubin, 2015).

---

[*]Corresponding author

37th Conference on Neural Information Processing Systems (NeurIPS 2023).

Many statistical tools have been developed to measure the dependence between random variables. Classical measures such as Pearson correlation (Pearson, 1920), Spearman's $\rho$ (Spearman, 1904) and Kendall's $\tau$ (Kendall, 1938) can only capture linear or monotonic dependence and may fail to detect complex associations. For univariate covariates, several rank-based measures have been proposed to overcome this limitation, such as those by Hoeffding (1948), Blum et al. (1961), Bergsma & Dassios (2014) and Chatterjee (2021). For multivariate covariates, various nonparametric tests have been introduced to quantify arbitrary associations, including the distance correlation (Székely et al., 2007), the projection correlation (Zhu et al., 2017), the mutual information (Berrett & Samworth, 2019), the kernel canonical correlation (Bach & Jordan, 2002), the constrained covariance (Gretton et al., 2005), and the Hilbert-Schmidt independence criterion (HSIC, Gretton et al., 2007). Among these measures, the distance correlation and the HSIC are perhaps the most widely used and studied in both statistics and machine learning. For instance, Li et al. (2012) applied the distance correlation to perform feature screening. Fan et al. (2020a) generalized the distance correlation to a conditional dependence measure under a factor model setting. Albert et al. (2022) developed an adaptive test of independence by aggregating HSIC measures. Pfister et al. (2018) defined a $d$-variable HSIC to characterize the mutual dependence among $d$ random variables. It is worth noting that the distance correlation is equivalent to HSIC with a specific kernel choice (Sejdinovic et al., 2013).

High-dimensional settings pose new challenges and opportunities for dependence measures. Székely & Rizzo (2013) suggested to estimate the distance correlation with the $U$-statistic theory to avoid bias accumulations when the dimension is high. Ramdas et al. (2015) found that the power of kernel and distance based independence tests decreases polynomially with increasing dimension against some fair alternatives. Zhu et al. (2020) proved that, when both $p$ and $q$ grow much faster than the sample size $n$, the sample distance correlation and HSIC can only detect componentwise linear dependences. Gao et al. (2021) established a general condition for the distance correlation based test to have power approaching 1. They also showed that the distance correlation can capture certain pure nonlinear relationship when $p = q = o(n^{1/2})$ under a specific alternative hypothesis.

In this paper, we aim to provide statistical insights into the HSIC in high dimensions, motivated by the increasing popularity of HSIC and the prevalence of high-dimensional data. We focus on the asymptotic distributions of HSIC under both null and alternative hypotheses when the dimensionality grows with the sample size. Under the null hypothesis, we prove that the rescaled HSIC converges in distribution to a standard normal distribution. We also derive a general condition for the HSIC based tests to have power asymptotically approaching one. These results extend those of Gao et al. (2021) on the distance correlation. Moreover, our analysis is more comprehensive than theirs. By decomposing the general condition, we reveal that HSIC can detect different types of dependences, depending on the dimensionality and sample orders. This sheds light on the performance of HSIC in high dimensions, which has been overlooked in the literature. For illustrative purposes, we let $\mathbf{x}$ and $\mathbf{y}$ be two random vectors with zero mean and identity covariance matrix. When HSIC is used to test for statistical independence between them, we obtain the following conditions for nontrivial power:

- If $p \to \infty$ and $q$ is fixed as $n \to \infty$, and there only exists the conditional mean of $\mathbf{x}$'s $s$-th polynomial given $\mathbf{y}$, then $p^{(s-1)}$ must grow slower than $n$.
- If both $p$ and $q \to \infty$ as $n \to \infty$, and there only exists the covariance between $\mathbf{x}$'s $s_1$-th polynomial and $\mathbf{y}$'s $s_2$-th polynomial, then $p^{(s_1-1)}q^{(s_2-1)}$ must grow slower than $n$.

These two conditions reveal significant statistical insights and have important implications in practice. For example, when the data dimension $q$ is small and $p$ is larger than the sample size $n$, according to the first condition, to ensure $p^{(s-1)}$ smaller than $n$, $s$ can only be 1. In other words, the HSIC can only measure the conditional mean of $\mathbf{x}$ given $\mathbf{y}$. When both $p$ and $q$ are larger than $n$, according to the second condition, to ensure $p^{(s_1-1)}q^{(s_2-1)}$ smaller than $n$, both $s_1$ and $s_2$ must be 1. That is, the HSIC can only measure the covariance between $\mathbf{x}$ and $\mathbf{y}$. In summary, our main contributions are:

1. We generalize the results of Gao et al. (2021) for distance correlation to HSIC, which is a more flexible and widely used measure of dependence.

2. We show that HSIC can capture different types of dependences, depending on the dimensionality and sample orders, which has rarely been realized in the literature before.

3. In contrast to Zhu et al. (2020), who only focused on the case when both $p$ and $q$ grow much faster than $n$, our results characterize a full picture of the performance of HSIC based test in high dimensions.

The rest of the paper is organized as follows. We give some preliminaries about the HSIC in Section 2. Then we present the asymptotic null distribution as well as the power analysis for high-dimensional HSIC in Section 3. In Section 4, we provide statistical insights into the HSIC through connecting the association type with the sample size, the high dimensionality, and the dependence structures within covariates. We conduct comprehensive numerical studies in Sections 5 and conclude this paper with a brief discussion in Section 6. All technical details are relegated to the Supplement.

## 2    Preliminaries

In this section, we give a brief review on HSIC measures proposed by Gretton et al. (2007), which is derived from the notion of cross-covariance operator Baker (1973). We assume that $X \in \mathcal{X} \subseteq \mathbb{R}^p$ and $Y \in \mathcal{Y} \subseteq \mathbb{R}^q$ are random vectors taking values in Euclidean spaces respectively. Let $\mathcal{F}$ and $\mathcal{G}$ be the reproducing kernel Hilbert spaces (RKHSs) on $\mathcal{X}$ and $\mathcal{Y}$, with associated kernels $K(\cdot, \cdot)$ and $L(\cdot, \cdot)$, respectively. Then the cross-covariance operator $C_{\mathbf{xy}}$ associated to $\mathcal{F}$ and $\mathcal{G}$ is defined as the operator mapping from $\mathcal{G}$ to $\mathcal{F}$, such that for all $f \in \mathcal{F}$ and $g \in \mathcal{G}$, $\langle f, C_{\mathbf{xy}} g \rangle_{\mathcal{F}} = \mathrm{cov}\{f(\mathbf{x}), g(\mathbf{y})\}$. Instead of using the largest singular value of this operator, Gretton et al. (2007) suggested to adopt the squared Hilbert-Schmidt norm, which admits a closed form expression

$$\begin{aligned} \mathrm{HSIC}(\mathbf{x}, \mathbf{y}) &= E\{K(\mathbf{x}_1, \mathbf{x}_2) L(\mathbf{y}_1, \mathbf{y}_2)\} + E\{K(\mathbf{x}_1, \mathbf{x}_2)\} E\{L(\mathbf{y}_1, \mathbf{y}_2)\} \\ &\quad - 2E\left[E\{K(\mathbf{x}_1, \mathbf{x}_2) \mid \mathbf{x}_1\} E\{L(\mathbf{y}_1, \mathbf{y}_2) \mid \mathbf{y}_1\}\right], \end{aligned}$$

where $(\mathbf{x}_1, \mathbf{y}_1)$ and $(\mathbf{x}_2, \mathbf{y}_2)$ are independent copies of $(\mathbf{x}, \mathbf{y})$. When the RKHSs, $\mathcal{F}$ and $\mathcal{G}$, are well chosen, (e.g., characteristic, Sriperumbudur et al., 2010), the HSIC can be served as an independence criterion in the sense that it is nonnegative and equals zero if and only if $\mathbf{x}$ and $\mathbf{y}$ are independent. To normalize the HSIC to range from zero to one, we follow Zhu et al. (2020) to define the squared Hilbert-Schmidt correlation as

$$\mathrm{hCorr}^2(\mathbf{x}, \mathbf{y}) \stackrel{\mathrm{def}}{=} \mathrm{HSIC}(\mathbf{x}, \mathbf{y}) \mathrm{HSIC}^{-1/2}(\mathbf{x}, \mathbf{x}) \mathrm{HSIC}^{-1/2}(\mathbf{y}, \mathbf{y}). \tag{1}$$

In addition, similar to Zhu et al. (2020) and Albert et al. (2022), we focus on the kernels that can be written in the form of $K(\mathbf{x}_1, \mathbf{x}_2) = k(\|\mathbf{x}_1 - \mathbf{x}_2\|/\gamma_{\mathbf{x}})$ and $L(\mathbf{y}_1, \mathbf{y}_2) = l(\|\mathbf{y}_1 - \mathbf{y}_2\|/\gamma_{\mathbf{y}})$, where $k(\cdot)$ and $l(\cdot)$ are some real-valued functions defined in $[0, \infty)$, $\gamma_{\mathbf{x}}$ and $\gamma_{\mathbf{y}}$ are the bandwidth parameters. This kind of kernel is very commonly used in the literature and includes many positive-definite kernel such as the Gaussian kernel, the Laplacian kernel, the rational quadratic kernel, and the Kernel generating Sobolev spaces, etc. For example, when $k(x) = \exp(-x^2)$, it corresponds to the Gaussian kernel, and when $k(x) = \exp(-x)$, it corresponds to the Laplacian kernel. One may refer to Genton (2001) for more examples and discussions about isotropic kernels.

## 3    Asymptotic Properties in High Dimensions

At the sample level, with the random sample $\{(\mathbf{x}_i, \mathbf{y}_i), i = 1, ..., n\}$, we estimate the HSIC with $U$-statistics,

$$\begin{aligned} \mathrm{HSIC}_n(\mathbf{x}, \mathbf{y}) &\stackrel{\mathrm{def}}{=} \frac{\sum_{(i_1, i_2)} K(\mathbf{x}_{i_1}, \mathbf{x}_{i_2}) L(\mathbf{y}_{i_1}, \mathbf{y}_{i_2})}{n(n-1)} - \frac{2 \sum_{(i_1, i_2, i_3)} K(\mathbf{x}_{i_1}, \mathbf{x}_{i_2}) L(\mathbf{y}_{i_1}, \mathbf{y}_{i_3})}{n(n-1)(n-2)} \\ &\quad + \frac{\sum_{(i_1, i_2, i_3, i_4)} K(\mathbf{x}_{i_1}, \mathbf{x}_{i_2}) L(\mathbf{y}_{i_3}, \mathbf{y}_{i_4})}{n(n-1)(n-2)(n-3)}, \end{aligned} \tag{2}$$

where the indexes in the summands, i.e., $(i_1, i_2)$, $(i_1, i_2, i_3)$ and $(i_1, i_2, i_3, i_4)$, are all distinctive from each other. Then the squared Hilbert-Schmidt correlation $\mathrm{hCorr}^2(\mathbf{x}, \mathbf{y})$ can be estimated as

$$\mathrm{hCorr}_n^2(\mathbf{x}, \mathbf{y}) \stackrel{\mathrm{def}}{=} \mathrm{HSIC}_n(\mathbf{x}, \mathbf{y}) \mathrm{HSIC}_n^{-1/2}(\mathbf{x}, \mathbf{x}) \mathrm{HSIC}_n^{-1/2}(\mathbf{y}, \mathbf{y}).$$

We now discuss how to use the sample squared Hilbert-Schmidt correlation to distinguish between the null hypothesis $H_0 : \mathbf{x}$ is independent of $\mathbf{y}$, and the alternative hypothesis $H_1 : \mathbf{x}$ is not independent of $\mathbf{y}$. To implement the HSIC based test, it is required to study the asymptotic distributions for $\mathrm{hCorr}_n^2(\mathbf{x}, \mathbf{y})$ under the null hypothesis. When the dimensions are fixed, Gretton et al. (2007) showed that $n\, \mathrm{hCorr}_n^2(\mathbf{x}, \mathbf{y})$ converges in distribution to a weighted sum of independent chi-squared random variables as long as $\mathbf{x}$ is independent of $\mathbf{y}$. We now study its asymptotic null distribution in high

dimensions. Before that, we define the two quantities $H_{\mathbf{x}}(\mathbf{x}_1, \mathbf{x}_2)$ and $H_{\mathbf{y}}(\mathbf{y}_1, \mathbf{y}_2)$ as

$$H_{\mathbf{x}}(\mathbf{x}_1, \mathbf{x}_2) \stackrel{\text{def}}{=} K(\mathbf{x}_1, \mathbf{x}_2) - E\{K(\mathbf{x}_1, \mathbf{x}_2) \mid \mathbf{x}_1\} - E\{K(\mathbf{x}_1, \mathbf{x}_2) \mid \mathbf{x}_2\} + E\{K(\mathbf{x}_1, \mathbf{x}_2)\},$$

$$H_{\mathbf{y}}(\mathbf{y}_1, \mathbf{y}_2) \stackrel{\text{def}}{=} L(\mathbf{y}_1, \mathbf{y}_2) - E\{L(\mathbf{y}_1, \mathbf{y}_2) \mid \mathbf{y}_1\} - E\{L(\mathbf{y}_1, \mathbf{y}_2) \mid \mathbf{y}_2\} + E\{L(\mathbf{y}_1, \mathbf{y}_2)\}. \quad (3)$$

We further define another two quantities $G_{\mathbf{x}}(\mathbf{x}_1, \mathbf{x}_2)$ and $G_{\mathbf{y}}(\mathbf{y}_1, \mathbf{y}_2)$ as

$$G_{\mathbf{x}}(\mathbf{x}_1, \mathbf{x}_2) \stackrel{\text{def}}{=} E\{H_{\mathbf{x}}(\mathbf{x}_1, \mathbf{x}_3)H_{\mathbf{x}}(\mathbf{x}_2, \mathbf{x}_3) \mid \mathbf{x}_1, \mathbf{x}_2\},$$

$$G_{\mathbf{y}}(\mathbf{y}_1, \mathbf{y}_2) \stackrel{\text{def}}{=} E\{H_{\mathbf{y}}(\mathbf{y}_1, \mathbf{y}_3)H_{\mathbf{y}}(\mathbf{y}_2, \mathbf{y}_3) \mid \mathbf{y}_1, \mathbf{y}_2\}.$$

Then we summarize the asymptotic null distribution for $\mathrm{hCorr}_n^2(\mathbf{x}, \mathbf{y})$ in the following Theorem.

**Theorem 1.** *Assume the kernels are symmetric with finite fourth moment, i.e., $K(\mathbf{x}_1, \mathbf{x}_2) = K(\mathbf{x}_2, \mathbf{x}_1)$, $L(\mathbf{y}_1, \mathbf{y}_2) = L(\mathbf{y}_2, \mathbf{y}_1)$, $E\{K^4(\mathbf{x}_1, \mathbf{x}_2)\} < \infty$ and $E\{L^4(\mathbf{y}_1, \mathbf{y}_2)\} < \infty$. Further assume that $p + q \to \infty$,*

$$\frac{E\{H_{\mathbf{x}}^4(\mathbf{x}_1, \mathbf{x}_2)\}E\{H_{\mathbf{y}}^4(\mathbf{y}_1, \mathbf{y}_2)\}}{n\{\mathrm{HSIC}(\mathbf{x}, \mathbf{x})\mathrm{HSIC}(\mathbf{y}, \mathbf{y})\}^2} \to 0, \quad \text{and} \quad \frac{E\{G_{\mathbf{x}}^2(\mathbf{x}_1, \mathbf{x}_2)\}E\{G_{\mathbf{y}}^2(\mathbf{y}_1, \mathbf{y}_2)\}}{\{\mathrm{HSIC}(\mathbf{x}, \mathbf{x})\mathrm{HSIC}(\mathbf{y}, \mathbf{y})\}^2} \to 0, \quad (4)$$

*as $n \to \infty$. Then under the null hypothesis, we have $2^{-1/2}n\,\mathrm{hCorr}_n^2(\mathbf{x}, \mathbf{y}) \xrightarrow{d} N(0, 1)$.*

Different from that when the dimensions are fixed, Theorem 1 reveals that, under certain conditions, $n\,\mathrm{hCorr}_n^2(\mathbf{x}, \mathbf{y})$ is asymptotically normal in high dimensions. The asymptotic normality makes the limiting null distribution tractable in practice. It greatly expedites the implementation of HSIC based tests because no additional permutations are required to decide critical values. We remark here that the assumptions imposed in Theorem 1 to ensure the asymptotic normality are generally very mild. For the restrictions on the kernels, many commonly used kernels, including Gaussian and Laplacian kernels, are symmetric with finite fourth moment. For the assumption in (4), it is used to restrict the weak dependence within covariates in $\mathbf{x}$ and $\mathbf{y}$. It holds true when $\mathbf{x}$ and $\mathbf{y}$ follow the multivariate normal distributions with bounded eigenvalues. It also holds true when $\mathbf{x}$ and $\mathbf{y}$ are $m$-dependent random sequences (Definition 9.1 of DasGupta, 2008). One may refer to Gao et al. (2021) for more discussions about this assumption.

Next, we explore the power performance of the HSIC based test in high dimensions. As discussed in Section 2, we restrict our attentions to the isotropic kernels. Specifically, in the remaining of this paper, we consider $K(\mathbf{x}_1, \mathbf{x}_2)$ and $L(\mathbf{y}_1, \mathbf{y}_2)$ to be in the form of $k(\|\mathbf{x}_1 - \mathbf{x}_2\|/\gamma_{\mathbf{x}})$ and $l(\|\mathbf{y}_1 - \mathbf{y}_2\|/\gamma_{\mathbf{y}})$, respectively. Because of the kernel form, we assume without loss of generality that $E\mathbf{x} = \mathbf{0}$ and $E\mathbf{y} = \mathbf{0}$. We denote by $\mathbf{z}^* \stackrel{\text{def}}{=} \mathbf{z}/\gamma_{\mathbf{z}}$, where $\mathbf{z} \in \mathbb{R}^d$ can be either $\mathbf{x}$ or $\mathbf{y}$. We say that $f(n) \asymp g(n)$ if $f(n) = O\{g(n)\}$ and $g(n) = O\{f(n)\}$. Before summarizing the asymptotic power performance, we make the following assumptions:

(A1) There exists some $\kappa_{\mathbf{z}} > 0$ such that $E\{\|\mathbf{z}^*\|^2 - E(\|\mathbf{z}^*\|^2)\}^{2k} \asymp E(\mathbf{z}_1^{*\mathrm{T}}\mathbf{z}_2^*)^{2k} \asymp d^{-k\kappa_{\mathbf{z}}}$ for all $k \in \mathbb{N}^+$.

(A2) Let $k_0(x) = k(x^{1/2})$ and $l_0(y) = l(y^{1/2})$. The first and second derivatives of $k_0(\cdot)$ and $l_0(\cdot)$ are uniformly bounded away from zero to infinity around $E\|\mathbf{x}_1^* - \mathbf{x}_2^*\|^2$ and $E\|\mathbf{y}_1^* - \mathbf{y}_2^*\|^2$, respectively.

We remark here that Assumption (A1) is used to restrict the dependence structures within the coordinates of $\mathbf{z}$. Because $\mathbf{z}^* = \mathbf{z}/\gamma_{\mathbf{z}}$ and $E\mathbf{z} = \mathbf{0}$, we have $E\mathbf{z}^* = \mathbf{0}$ as well. Hence, both $\|\mathbf{z}^*\|^2 - E(\|\mathbf{z}^*\|^2)$ and $\mathbf{z}_1^{*\mathrm{T}}\mathbf{z}_2^*$ are sums of $d$ random variables with mean zero. Their standardized versions may converge in distribution to standard normal distributions under mild conditions (e.g., mixing conditions) by applying the central limit theorem (CLT). For instance, Volnỳ (1989) established a CLT for non-stationary mixing processes. Alternatively, there exists some $\kappa_z > 0$ such that $d^{\kappa_z/2}\{\|\mathbf{z}^*\|^2 - E(\|\mathbf{z}^*\|^2)\}$ and $d^{\kappa_z/2}\mathbf{z}_1^{*\mathrm{T}}\mathbf{z}_2^*$ both converge in distribution to normal random variables. Then Condition (A1) is satisfied. Assumption (A2) imposes some conditions on the kernels, which covers the Gaussian and Laplacian kernels. Moreover, it also makes several requirements on the bandwidth parameters as well. For example, suppose the Gaussian kernel is used, i.e., $k_0(z) = \exp(-z)$, when $\gamma_{\mathbf{z}}$ is small enough such that $E\|\mathbf{z}_1^* - \mathbf{z}_2^*\|^2 \to \infty$, then we have $k_0'(E\|\mathbf{z}_1^* - \mathbf{z}_2^*\|^2) \to 0$, which violates Assumption (A2). Theoretically, the bandwidth parameter $\gamma_{\mathbf{z}}$ can be chosen from a wide range of values, as long as it satisfies condition (A2). In practice, we use the median of

$\|\mathbf{z}_1 - \mathbf{z}_2\|$ as a default value for $\gamma_{\mathbf{z}}$. When it has the same magnitude with $E\|\mathbf{z}_1 - \mathbf{z}_2\|$, Assumption (A2) can be easily satisfied for many commonly used kernels. Moreover, as demonstrated in the simulated examples in the Supplementary Material, our method exhibits robustness across various selections of $\gamma_{\mathbf{z}}$.

With these two assumptions, we are ready to provide the asymptotic power performance of the HSIC based test in high dimensions.

**Theorem 2.** *Assume (A1) and (A2) hold true. Then under the alternative hypothesis, if* $n^{1/2}\mathrm{hCorr}^2(\mathbf{x}, \mathbf{y}) \to \infty$ *as* $n \to \infty$, *we have* $n\,\mathrm{hCorr}_n^2(\mathbf{x}, \mathbf{y}) \to \infty$ *in probability.*

Theorem 2 reveals that, as long as the dependence measured by $\mathrm{hCorr}^2(\mathbf{x}, \mathbf{y})$ is not too small, $n\,\mathrm{hCorr}_n^2(\mathbf{x}, \mathbf{y})$ converges to infinity in probability. This, together with Theorem 1, guarantee that the HSIC based test can have nontrivial power in high dimensions. However, as the dimension increases, the signal measured by $\mathrm{hCorr}^2(\mathbf{x}, \mathbf{y})$ may decay to zero, which makes the test lose power in high dimensions. To gain more insights about how the power is influenced by the dimensions, it is desired to connect $\mathrm{hCorr}^2(\mathbf{x}, \mathbf{y})$ with the dimensionality, as well as the association types between $\mathbf{x}$ and $\mathbf{y}$.

## 4 Statistical Insights in High Dimensions

We have discussed the asymptotic properties of HSIC based tests in high dimensions, and demonstrated that it can detect the alternative hypothesis as long as the signal strength measured by $\mathrm{hCorr}^2(\mathbf{x}, \mathbf{y})$ does not decay to zero too fast. However, it is still unclear that how $\mathrm{hCorr}^2(\mathbf{x}, \mathbf{y})$ is influenced by the dimensions. To gain more insights into the performance in high dimensions, in this section, we study what kind of association types the HSIC can detect under different relationships between the sample size and the covariate dimensions. To this end, we expand $\mathrm{hCorr}^2(\mathbf{x}, \mathbf{y})$ defined in (1) to gain some insights. Because $\mathrm{HSIC}(\mathbf{x}, \mathbf{y})$ characterizes the dependence between $\mathbf{x}$ and $\mathbf{y}$, while $\mathrm{HSIC}(\mathbf{x}, \mathbf{x})$ and $\mathrm{HSIC}(\mathbf{y}, \mathbf{y})$ are the normalizing factors, we expand $\mathrm{HSIC}(\mathbf{x}, \mathbf{y})$ and calculate the orders for $\mathrm{HSIC}(\mathbf{x}, \mathbf{x})$ and $\mathrm{HSIC}(\mathbf{y}, \mathbf{y})$.

We first calculate the orders for $\mathrm{HSIC}(\mathbf{x}, \mathbf{x})$ and $\mathrm{HSIC}(\mathbf{y}, \mathbf{y})$ at the population level as the dimensions diverge to infinity. The results are summarized in the following Proposition.

**Proposition 1.** *Under Assumptions (A1) and (A2),* $\mathrm{HSIC}(\mathbf{z}, \mathbf{z}) \asymp d^{-\kappa_{\mathbf{z}}}$ *if* $d \to \infty$.

According to Proposition 1, the condition $n^{1/2}\mathrm{hCorr}^2(\mathbf{x}, \mathbf{y}) \to \infty$, which guarantees the HSIC based test would have power approaching 1 in high dimensions, will boil down to $n^{1/2}p^{\kappa_{\mathbf{x}}/2}q^{\kappa_{\mathbf{y}}/2}\mathrm{HSIC}(\mathbf{x}, \mathbf{y}) \to \infty$.

Next, we expand $\mathrm{HSIC}(\mathbf{x}, \mathbf{y})$ at the population level. Before that, we introduce some additional notations. Similar to Shao & Zhang (2014), we define $\mathrm{MD}^2(\mathbf{x} \mid \mathbf{y})$ as

$$\mathrm{MD}^2(\mathbf{x} \mid \mathbf{y}) \stackrel{\text{def}}{=} E\{(\mathbf{x}_1 - E\mathbf{x})^{\mathrm{T}}(\mathbf{x}_2 - E\mathbf{x})L(\mathbf{y}_1, \mathbf{y}_2)\},$$

where $(\mathbf{x}_1, \mathbf{y}_1)$ and $(\mathbf{x}_2, \mathbf{y}_2)$ are independent copies of $(\mathbf{x}, \mathbf{y})$. We show in Lemma 1 that, $\mathrm{MD}^2(\mathbf{x} \mid \mathbf{y}) = 0$ if and only if $E(\mathbf{x} \mid \mathbf{y}) = 0$. That is, $\mathrm{MD}^2(\mathbf{x} \mid \mathbf{y})$ measures the degree of conditional mean of $\mathbf{x}$ given $\mathbf{y}$, which quantifies the difference between $E(\mathbf{x} \mid \mathbf{y})$ and $E\mathbf{x}$.

**Lemma 1.** *Assume the kernel* $L(\mathbf{y}_1, \mathbf{y}_2) = l(\|\mathbf{y}_1 - \mathbf{y}_2\|/\gamma_{\mathbf{y}})$ *is a characteristic, bounded continuous, and real-valued positive definite function, then* $\mathrm{MD}^2(\mathbf{x} \mid \mathbf{y}) = 0$ *if and only if* $E(\mathbf{x} \mid \mathbf{y}) = 0$ *almost surely.*

We remark here that characteristic condition in Lemma 1 is used to guarantee that the HSIC is nonnegative and equals zero if and only if $\mathbf{x}$ and $\mathbf{y}$ are independent. One may refer to (Sriperumbudur et al., 2010) for more details about this condition. Furthermore, we let $k_0^{(i)}$ be the $i$-th derivative of $k_0(\cdot)$ evaluated at $E\|\mathbf{x}_1^* - \mathbf{x}_2^*\|^2$, and $l_0^{(j)}$ is the $j$-th derivative of $l_0(\cdot)$ evaluated at $E\|\mathbf{y}_1^* - \mathbf{y}_2^*\|^2$. Let $\mathbf{x}^{\otimes n}$ be the $n$-th kronecker power of $\mathbf{x}$, which is defined as $\mathbf{x}^{\otimes n} = \mathbf{x} \otimes \mathbf{x}^{\otimes(n-1)}$, $\mathbf{x}^{\otimes 1} = \mathbf{x}$, and $\otimes$ denotes the Kronecker product. In addition, we use $\|\cdot\|_F$ to represent the Frobenius norm of a matrix. Then we expand $\mathrm{HSIC}(\mathbf{x}, \mathbf{y})$ in the following Theorem.

**Theorem 3.** *Assume (A1) and (A2) hold true. Then under the alternative hypothesis,*

1. *when $p \to \infty$ and $q$ remains fixed as $n \to \infty$, if $E(\mathbf{x}^{\otimes t} \mid \mathbf{y}) = E(\mathbf{x}^{\otimes t})$ hold true for all $t < s$ for some $s \in \mathbb{N}^+$, then $\mathrm{HSIC}(\mathbf{x}, \mathbf{y}) = O(p^{-s\kappa_{\mathbf{x}}/2})$, and*

$$\mathrm{HSIC}(\mathbf{x}, \mathbf{y}) = k_0^{(s)} \sum_{2a+c=s} \frac{(-2)^c}{a!a!c!} \mathrm{MD}^2(\mathbf{x}^{*\otimes c}\|\mathbf{x}^*\|^{2a} \mid \mathbf{y}) + o(p^{-s\kappa_{\mathbf{x}}/2}),$$

2. *when $p \to \infty$ and $q \to \infty$ as $n \to \infty$, if $\mathrm{cov}(\mathbf{x}^{\otimes t_1}, \mathbf{y}^{\otimes t_2}) \neq \mathbf{0}$ only when $t_1 \geq s_1$ and $t_2 \geq s_2$ for some $s_1, s_2 \in \mathbb{N}^+$, then $\mathrm{HSIC}(\mathbf{x}, \mathbf{y}) = O(p^{-s_1\kappa_{\mathbf{x}}/2}q^{-s_2\kappa_{\mathbf{y}}/2})$, and*

$$\mathrm{HSIC}(\mathbf{x}, \mathbf{y}) = \sum_{2a_1+c_1=s_1} \sum_{2a_2+c_2=s_2} \frac{k_0^{(s_1)}(-2)^{c_1}}{a_1!a_1!c_1!} \frac{l_0^{(s_2)}(-2)^{c_2}}{a_2!a_2!c_2!} \left\| \mathrm{cov}\{\|\mathbf{x}^*\|^{2a_1}\mathbf{x}^{*\otimes c_1}, \|\mathbf{y}^*\|^{2a_2}\mathbf{y}^{*\otimes c_2 T}\} \right\|_F^2$$
$$+ o(p^{-s_1\kappa_{\mathbf{x}}/2}q^{-s_2\kappa_{\mathbf{y}}/2}).$$

Theorem 3 enables us to gain some insights into the HSIC in high dimensions. First of all, because when $2a + c = s$, $\mathbf{x}^{*\otimes c}\|\mathbf{x}^*\|^{2a}$ is the $s$-th polynomial for $\mathbf{x}$, and $\mathrm{MD}^2(\mathbf{x}^{*\otimes c}\|\mathbf{x}^*\|^{2a} \mid \mathbf{y})$ measures the departure from the conditional mean independence of $\mathbf{x}$ given $\mathbf{y}$. Then the first part of Theorem 3 implies that when $p \to \infty$ and $q$ remains fixed as $n \to \infty$, if $E(\mathbf{x}^{\otimes t} \mid \mathbf{y}) = E(\mathbf{x}^{\otimes t})$ hold true for all $t < s$ for some $s \in \mathbb{N}^+$, the HSIC quantifies the departure from $E(\mathbf{x}^{\otimes s} \mid \mathbf{y}) = E(\mathbf{x}^{\otimes s})$. In other words, when only one dimension of the covariates diverges to infinity, the HSIC first measures the conditional mean of $\mathbf{x}$ given $\mathbf{y}$. If there is no such mean dependence, it turns to measure the conditional mean of $\mathbf{x}^{\otimes 2}$ given $\mathbf{y}$, etc. Similarly, when $p \to \infty$ and $q \to \infty$ as $n \to \infty$, the second part of Theorem 3 reveals that, the HSIC may first search for the covariance between $\mathbf{x}$ given $\mathbf{y}$. If such covariance equals zero, it then turns to search for the covariance between $\mathbf{x}^{\otimes 2}$ and $\mathbf{y}$, or the covariance between $\mathbf{x}$ and $\mathbf{y}^{\otimes 2}$, etc. Therefore, Theorem 3 characterizes the changing process of the ability to measure nonlinear dependence in high dimensions for HSIC. As a comparison, Zhu et al. (2020) only showed that, when both dimensions diverge much faster than the sample size, the HSIC can only capture the marginal linear dependence. From this point of view, our result is a generalization of theirs.

When the HSIC is used to test for statistical independence, Theorem 2 and Proposition 1 ensure that, when $n^{1/2}p^{\kappa_{\mathbf{x}}/2}q^{\kappa_{\mathbf{y}}/2}\mathrm{HSIC}(\mathbf{x}, \mathbf{y}) \to \infty$, the HSIC based test would have asymptotic power 1. To make this condition hold true, the first part of Theorem 3 implies that when $p \to \infty$ and $q$ remains fixed as $n \to \infty$, if there only exists the conditional mean of $\mathbf{x}^{\otimes s}$ given $\mathbf{y}$, it is required that $p^{(s-1)\kappa_{\mathbf{x}}} = o(n)$ because $\mathrm{HSIC}(\mathbf{x}, \mathbf{y})$ is of order $O(p^{-s\kappa_{\mathbf{x}}/2})$ in this case. In other words, when $n = O\{p^{(s-1)\kappa_{\mathbf{x}}}\}$, the HSIC based test can have nontrivial power only when there exists the $i$-th polynomial mean dependence of $\mathbf{x}$ given $\mathbf{y}$ for some $i < s$. Similarly, when both $p$ and $q$ diverges as $n \to \infty$, the second part of Theorem 3, when there only exists the covariance between $\mathbf{x}^{\otimes s_1}$ and $\mathbf{y}^{\otimes s_2}$, the test based on HSIC can have nontrivial power only when $p^{(s_1-1)\kappa_{\mathbf{x}}}q^{(s_2-1)\kappa_{\mathbf{y}}} = o(n)$. Because $\kappa_{\mathbf{x}}$ represents the degree of dependence within covariates, Theorem 3 successfully connects the association type between $\mathbf{x}$ and $\mathbf{y}$, with the sample size, the high dimensionality, and the dependence structures within covariates. We remark here that Gao et al. (2021) only justified that the test based on distance correlation can have power approaching 1 when $p = q = o(n^{1/2})$ if there exist a pure nonlinear dependence between $\mathbf{x}$ and $\mathbf{y}$. Therefore, their result is in spirit a special case of ours.

## 5 Numerical Studies

### 5.1 Simulations

In this subsection, we conduct some simulation studies to validate our theoretical conclusions on the HSIC based test in high dimensions. In particular, in Example 1, we show that, as long as the dimensions are not very small, the normal approximation can approximate the null distribution quite well, although it requires the dimension to diverge to infinity in theory. In Example 2, we fix the dimension of $\mathbf{y}$ and increase the dimension of $\mathbf{x}$, and investigate the finite sample power performance of the HSIC based tests under various mean dependence types. In Example 3, we increase the dimensions of $\mathbf{x}$ and $\mathbf{y}$ simultaneously, and inspect how the empirical power is influenced by the covariate dimensions under different relationships.

Throughout the simulations, we choose two kinds of commonly used kernels to implement the HSIC based tests, i.e., Gaussian and Laplacian. For the Gaussian kernel, we set $K(\mathbf{x}_1, \mathbf{x}_2) =$

$\exp\{-\|\mathbf{x}_1 - \mathbf{x}_2\|^2/(2\gamma_{\mathbf{x}}^2)\}$ and $L(\mathbf{y}_1, \mathbf{y}_2) = \exp\{-\|\mathbf{y}_1 - \mathbf{y}_2\|^2/(2\gamma_{\mathbf{y}}^2)\}$. While for the Laplacian kernel, we set $L(\mathbf{x}_1, \mathbf{x}_2) = \exp(-\|\mathbf{x}_1 - \mathbf{x}_2\|/\gamma_{\mathbf{x}})$ and $L(\mathbf{y}_1, \mathbf{y}_2) = \exp(-\|\mathbf{y}_1 - \mathbf{y}_2\|/\gamma_{\mathbf{y}})$. For both choices of kernels, we set the bandwidth parameters to be $\gamma_{\mathbf{x}} = c_0\gamma_{\mathbf{x}}^m$ and $\gamma_{\mathbf{y}} = c_0\gamma_{\mathbf{y}}^m$ and vary $c_0$ from 0.5 to 2. Here, $\gamma_{\mathbf{z}}^m$ represents the median of $\{\|\mathbf{z}_i - \mathbf{z}_j\|\}_{1 \le i < j \le n}$, and $\mathbf{z}$ is either $\mathbf{x}$ or $\mathbf{y}$. Due to space constraints, we present results specifically for the case when $\gamma_{\mathbf{x}} = \gamma_{\mathbf{x}}^m$ and $\gamma_{\mathbf{y}} = \gamma_{\mathbf{y}}^m$. The remaining results are summarized in the Supplementary Material. Notably, our findings remain consistent across different values of $\gamma_{\mathbf{x}}$ and $\gamma_{\mathbf{y}}$, showing the stability and reliability of our approach.

**Example 1**. In this example, we examine the normal approximation accuracy under the null hypothesis. Towards this goal, we simply generate $\mathbf{x} = (X_1, \ldots, X_p)^{\mathrm{T}} \in \mathbb{R}^p$ from a multivariate normal distribution with mean zero and covariance matrix $\mathbf{\Sigma}_{\mathbf{x}} = \mathbf{I}_{p \times p}$. Then we independently generate $\mathbf{y} = (Y_1, \ldots, Y_q)^{\mathrm{T}} \in \mathbb{R}^q$ from another a multivariate normal distribution whose coordinates follow iid standard normal distributions. We fix the sample size to be 100 and consider two scenarios for the covariate dimensions. In the first scenario, we fix $q = 1$ and vary $p$ from $\{5, 25, 100\}$. In the second scenario, we set $p = q = d$ and vary $d$ from $\{2, 5, 10\}$. For each setting, we repeat the experiments 5000 times and compare the empirical null distributions of the test statistics, $\{2^{-1/2}n\,\mathrm{hCorr}_n^2(\mathbf{x}, \mathbf{y})\}$, with the standard normal distribution. Specifically, we display the kernel density curves of the test statistics for the two scenarios in Figures 1 and 2, respectively.

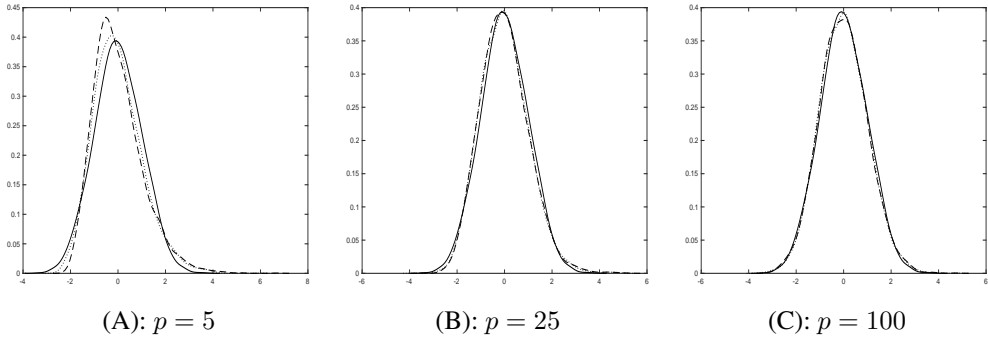

|  |  |  |
|---|---|---|
| (A): $p = 5$ | (B): $p = 25$ | (C): $p = 100$ |

Figure 1: The kernel densities of the test statistics under the null hypothesis computed from 5000 simulations. We fix $q = 1$ and vary $p$ from $\{5, 25, 100\}$. The horizontal axes represent the observed values of the test statistics, and the vertical axes represent the kernel densities of those values. We choose two different kernels to implement the tests, i.e., Gaussian (dashed line) and Laplacian (dotted line). The solid line is the reference curve, which is the density of the standard normal distribution.

According to Figures 1 and 2, we can see that, when the dimensions are small, the normal distribution cannot approximate the null distribution well. For example, when $p = q = 2$, the empirical null distributions for the tests with both Gaussian and Laplacian kernels are quite distinct from the reference curve in Figure 2 (A). However, as the dimensions increase, the accuracy of the normal approximation for the null distribution becomes quite precise, regardless of the choice of the kernels. For example, in both Figures 1 and 2, the empirical null distributions are almost the same as the reference curve as long as $pq$ are not smaller than 25.

**Example 2**. In this example, we investigate how the empirical power is influenced by the dimension when only one of the covariate dimensions increases. For simplicity, we fix the sample size to be $n = 100$, as that in Example 1. As for the covariate dimensions, we fix $q = 1$ and vary $p$ from $\{30, 50, 100, 200, 500, 1000\}$. We consider three models to inspect the power performances. For Models (I) and (II), we generate $\mathbf{x}$ from a multivariate normal distribution with mean zero and covariance matrix $\mathbf{\Sigma}_{\mathbf{x}} = (0.5^{|i-j|})_{p \times p}$. Then we generate an independent error term $\varepsilon$ that follows standard normal distribution. The univariate response $Y$ is generated through

$$\text{Model (I)}: \quad Y = X_1 + \ldots + X_p + \varepsilon;$$
$$\text{Model (II)}: \quad Y = X_1^2 + \ldots + X_p^2 + \varepsilon.$$

For Model (III), we first generate $Y \sim N(0, 1)$. Conditional on $Y$, we generate $\mathbf{x}$ through

$$\text{Model (III)}: \quad \{(X_{2k-1}, X_{2k})^{\mathrm{T}} \mid Y\} \sim N\{\mathbf{0}, (\rho_{k,Y}^{|i-j|})_{2 \times 2}\}, \quad k = 1, \ldots, p/2,$$

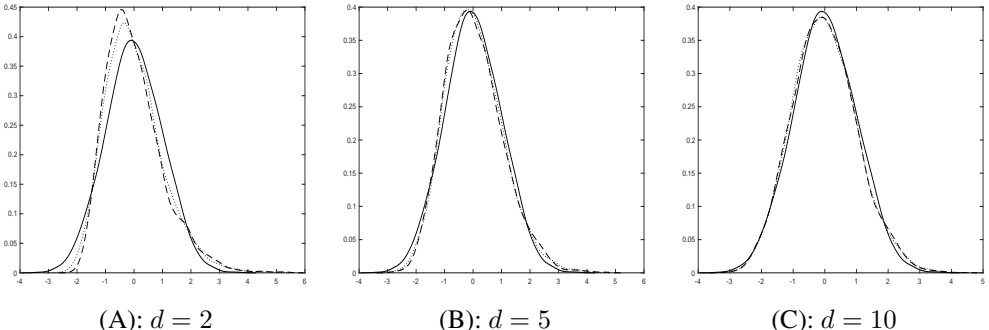

| (A): $d = 2$ | (B): $d = 5$ | (C): $d = 10$ |

Figure 2: The kernel densities of the test statistics under the null hypothesis computed from 5000 simulations. We set $p = q = d$ and vary $d$ from $\{2, 5, 10\}$. The horizontal axes represent the observed values of the test statistics, and the vertical axes represent the kernel densities of those values. We choose two different kernels to implement the tests, i.e., Gaussian (dashed line) and Laplacian (dotted line). The solid line is the reference curve, which is the density of the standard normal distribution.

Table 1: The empirical powers of different tests for Models (I)-(III) in Example 2. The significance level is 0.05. We fix $q = 1$ and vary $p$ from $\{30, 50, 100, 200, 500, 1000\}$.

| Model | Test | $p$ | | | | | |
| | | 30 | 50 | 100 | 200 | 500 | 1000 |
|---|---|---|---|---|---|---|---|
| | Gaussian | 1.000 | 1.000 | 1.000 | 1.000 | 0.998 | 0.954 |
| (I) | Laplacian | 1.000 | 1.000 | 1.000 | 1.000 | 0.994 | 0.916 |
| | DC | 1.000 | 1.000 | 1.000 | 1.000 | 1.000 | 0.998 |
| | Gaussian | 0.934 | 0.774 | 0.484 | 0.318 | 0.184 | 0.132 |
| (II) | Laplacian | 0.998 | 0.978 | 0.834 | 0.596 | 0.314 | 0.234 |
| | DC | 0.974 | 0.894 | 0.650 | 0.412 | 0.226 | 0.176 |
| | Gaussian | 0.044 | 0.050 | 0.060 | 0.046 | 0.060 | 0.068 |
| (III) | Laplacian | 0.050 | 0.046 | 0.054 | 0.048 | 0.054 | 0.060 |
| | DC | 0.050 | 0.050 | 0.052 | 0.034 | 0.064 | 0.044 |

where $\rho_{k,Y} = (2\delta_k - 1)\Phi(Y)$, $\delta_k$s are iid Bernoulli(0.5) random variables, and $\Phi(\cdot)$ is the cumulative distribution function of a standard normal distribution. In addition, $(X_{2k-1}, X_{2k})^{\mathrm{T}}$s are mutually independent conditional on $Y$. We remark here that in Model (III), $X_k$ is marginally independent of $Y$ for all $k = 1, \ldots, p$. It can be verified that there exist linear, quadratic and quartic conditional means of $\mathbf{x}$ given $Y$ in Models (I), (II) and (III), respectively.

As we have demonstrated in Example 1, when $q = 1$, the standard normal distribution can approximate the null distribution pretty well as long as $p$ is no less than 25, we use the standard normal distribution to obtain $p$-values. We repeat each experiment 500 times and report the empirical powers in Table 1. As a comparison, we also include the results for the distance correlation in Table 1. We can observe that all the empirical powers for Model (I) can still approach one, implying that the linear mean dependence can be easily detected, even when the dimension is much higher than the sample size. For example, when $p = 1000$, the power for Model (I) when the Gaussian kernel is used is still as high as 0.954. However, when there only exists some higher order of mean dependence, the empirical powers drop significantly. In addition, the performances deteriorate as the order of mean dependence increase. For example, when $p = 100$ and we choose the Gaussian kernel to implement the test, the empirical power for Model (III) is only 0.060, which is smaller than 0.484, the empirical power for Model (II).

**Example 3**. In this example, we study how the dimensionality can effect the empirical power when both of the covariate dimensions increase simultaneously. We also fix the sample size to be $n = 100$. We set $p = q = d$ and vary $d$ from $\{6, 10, 20, 50, 100, 200\}$. Similar to Example 2, we consider three models in this example. For Models (IV) and (V), we generate $\mathbf{x}$ the same as that in Models (I) and (II), i.e., $\mathbf{x} \sim N\{\mathbf{0}, (0.5^{|i-j|})_{p \times p}\}$. The independent error terms $\varepsilon_1, \ldots, \varepsilon_d$ are generated from $d$ independent standard normal distributions. Then we generate $\mathbf{y} = (Y_1, \ldots, Y_d)^{\mathrm{T}}$ through

$$\text{Model (IV)}: \quad Y_j = X_j + \varepsilon_j, \quad j = 1, \ldots, d;$$

Table 2: The empirical powers of different tests for Models (IV)-(VI) in Example 3. The significance level is 0.05. We set $p = q = d$ and vary $d$ from $\{6, 10, 20, 50, 100, 200\}$.

| Model | Test | $d$ | | | | | |
|---|---|---|---|---|---|---|---|
| | | 6 | 10 | 20 | 50 | 100 | 200 |
| (IV) | Gaussian | 1.000 | 1.000 | 1.000 | 1.000 | 1.000 | 1.000 |
| | Laplacian | 1.000 | 1.000 | 1.000 | 1.000 | 1.000 | 1.000 |
| | DC | 1.000 | 1.000 | 1.000 | 1.000 | 1.000 | 1.000 |
| (V) | Gaussian | 1.000 | 1.000 | 0.944 | 0.440 | 0.242 | 0.140 |
| | Laplacian | 1.000 | 1.000 | 1.000 | 0.904 | 0.578 | 0.282 |
| | DC | 1.000 | 0.986 | 0.844 | 0.400 | 0.232 | 0.136 |
| (VI) | Gaussian | 0.056 | 0.064 | 0.046 | 0.078 | 0.038 | 0.072 |
| | Laplacian | 0.058 | 0.064 | 0.044 | 0.074 | 0.040 | 0.072 |
| | DC | 0.060 | 0.066 | 0.046 | 0.078 | 0.040 | 0.072 |

$$\text{Model (V)}: \qquad Y_j = X_j^2 + \varepsilon_j, \quad j = 1, \ldots, d.$$

For Model (VI), we first generate all coordinates of $\mathbf{y}$ independently from standard normal distributions. Then given $\mathbf{y}$, we generate $\mathbf{x}$ through

$$\text{Model (VI)}: \quad \{(X_{2k-1}, X_{2k})^{\mathrm{T}} \mid \mathbf{y}\} \sim N\{\mathbf{0}, (\rho_{k,\mathbf{y}}^{|i-j|})_{2\times 2}\}, \quad k = 1, \ldots, d/2,$$

where $\rho_{k,\mathbf{y}} = (2\delta_k - 1)\Phi(Y_{2k})$ and $\delta_k$s are iid Bernoulli(0.5) random variables. Meanwhile, we generate $(X_{2k-1}, X_{2k})^{\mathrm{T}}$s independently conditional on $\mathbf{y}$. Similar to Models (I), (II) and (III), there exist covariances between $\mathbf{y}$ and linear, quadratic and quartic polynomials of $\mathbf{x}$ in Models (IV), (V) and (VI), respectively.

Following that in Example 2, we use the standard normal distribution to obtain $p$-values and repeat each experiment 500 times to report the empirical powers. The corresponding results are summarized in Table 2, from which we observe a similar phenomenon. That is, the empirical powers decay as the dimensions increase, and the decay rates increase when there only exist higher orders of covariances between $\mathbf{x}$ and $\mathbf{y}$. In addition, according to both Tables 1 and 2, the conclusions for the HSIC with Gaussian and Laplacian kernels also apply to the distance correlation. This is in line with our anticipation because the distance correlation is equivalent to the HSIC with some particular kernel.

## 5.2 Real Data Application

The prices of gasoline and many other fuels may have important impacts on raw materials and related companies. For example, the petroleum is widely used as upstream raw materials by many chemical industries. We are interested in whether there is any dependency between stock prices of the energy sector and the raw material sector in the U.S. stock market. To see this, we extract the monthly mean stock prices of energy companies as well as raw material companies starting from January 2021 to December 2022 from https://finance.yahoo.com/. According to the Global Industry Classification Standard, we get 345 and 294 companies from raw material and energy sectors, respectively. We remove those stocks whose data are not complete in this period, yielding 224 companies from the raw material sector and 214 companies from the energy sector, respectively. At each month $t$, $1 \leq t \leq 24$, we denote the monthly mean stock prices of these companies from the two sectors by $\mathbf{x}_t = (X_{t1}, \ldots, X_{tp})^{\mathrm{T}}$ and $\mathbf{y}_t = (Y_{t1}, \ldots, Y_{tq})^{\mathrm{T}}$, where $p = 224$ and $q = 214$. Let $S_{ti}^{\mathbf{x}} = \log(X_{ti}/X_{t-1,i})$ and $S_{tj}^{\mathbf{y}} = \log(Y_{tj}/Y_{t-1,j})$ be the stock returns at month $t$ for the $i$-th company in the raw material sector, and $j$-th company in the energy sector, respectively. We test whether the stock returns in these two sectors are independent using $\{(\mathbf{s}_t^{\mathbf{x}}, \mathbf{s}_t^{\mathbf{y}})\}_{t=2}^{24}$, where $\mathbf{s}_t^{\mathbf{x}} = (S_{t1}^{\mathbf{x}}, \ldots, S_{tp}^{\mathbf{x}})^{\mathrm{T}}$ and $\mathbf{s}_t^{\mathbf{y}} = (S_{t1}^{\mathbf{y}}, \ldots, S_{tq}^{\mathbf{y}})^{\mathrm{T}}$. We implement the test using HSIC with Gaussian and Laplacian kernels. The resulting $p$-values are $2.031 \times 10^{-10}$ and $2.749 \times 10^{-9}$, respectively. Then we can conclude that there exists dependences between the monthly mean stock prices of the energy sector and the raw material sector.

We also compare the RV coefficient with our proposed method in this dataset. The RV coefficient yields a $p$-value of $2.02 \times 10^{-4}$, which indicates the presence of linear dependences in this dataset. We remark that, in this data set, both covariates dimensions $p$ and $q$ are much larger than sample size $n$. According to the second part of Theorem 3 and the discussions at the end of Section 4, the HSIC can only have nontrivial power if $p^{(s_1-1)\kappa_{\mathbf{x}}}q^{(s_2-1)\kappa_{\mathbf{y}}} = o(n)$. In the context of our dataset, this

condition is satisfied exclusively when $s_1 = s_2 = 1$, signifying a covariance relationship between $\mathbf{x}$ and $\mathbf{y}$. This together with the fact that the test based on HSIC rejected the null hypothesis, we conclude that there exists a linear dependence relationship between $\mathbf{x}$ and $\mathbf{y}$. This is consistent with the RV coefficient result. Moreover, we identify that there is a strong linear relationship between stock returns for two pairs of companies: Denison Mines Corp. and Energy Fuels Inc., and Uranium Energy Corp. and Energy Fuels Inc. The $R^2$s for the corresponding linear models are 0.8779 and 0.8718, respectively. This means that the models fit the data very well and that the stock returns move together in a predictable way.

## 6 Conclusion and Discussion

This paper aims to investigate the performance of HSIC in high dimensions from a theoretical perspective. We first prove that the asymptotic null distribution of a rescaled HSIC is a standard normal in the high dimensional setting. Then we derive a general condition for the HSIC based tests to have power asymptotically approaching one. To gain more insights, we decompose this condition and show that it depends on the sample size, the covariate dimensions, the dependence structures within covariates, and the association types between $\mathbf{x}$ and $\mathbf{y}$. We also show that the ability of HSIC to detect nonlinear dependence deteriorates as the dimensions increase. Most existing works focused on either fixed dimensions or dimensions that diverge much faster than the sample size. We fill this gap in the literature and provide a comprehensive picture of the performance of HSIC based tests.

Our theory also reveals how the ability of HSIC to capture nonlinear dependence diminishes in high dimensions. To improve the power performance, Zhu et al. (2020) proposed to aggregate marginal sample HSIC as the test statistic instead of using HSIC over the whole features. However, this approach only measures marginal dependences, and it is well known that marginal independence does not imply joint independence. Moreover, because it sums over all pairs of marginal HSIC, it may suffer from a significant power loss when the alternative is sparse. Developing an independence test that has nontrivial power against all kinds of dependences remains an open question. Therefore, it would be interesting to explore new methods that can enhance power performances in high dimensions.

## Acknowledgments and Disclosure of Funding

The authors thank five anonymous reviewers for their comments and suggestions that significantly enhanced the quality of the paper. All authors contribute equally to this paper and their names are listed in an alphabetical order. This work is supported by National Natural Science Foundation of China (12271331 and 72192832), and Program for Innovative Research Team of Shanghai University of Finance and Economics.

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
