# Supplement to "Statistical Insights into HSIC in High Dimensions"

**Tao Zhang**
School of Information Management and Engineering
Shanghai University of Finance and Economics
Shanghai 200433, China
dalizhangt@163.com

**Yaowu Zhang**[*]
School of Information Management and Engineering
Shanghai University of Finance and Economics
Shanghai 200433, China
zhang.yaowu@mail.shufe.edu.cn

**Tingyou Zhou**
School of Data Sciences
Zhejiang University of Finance and Economics
Hangzhou 310018, China
zhoutingyou@zufe.edu.cn

## S.1 Additional Simulations about bandwidth parameters

In this section, we verify the robustness of our method in practical scenarios with various bandwidth parameter choices. In particular, we set the bandwidth parameters to be $\gamma_{\mathbf{x}} = c_0 \gamma_{\mathbf{x}}^m$ and $\gamma_{\mathbf{y}} = c_0 \gamma_{\mathbf{y}}^m$ and vary $c_0$ from 0.5 to 2. Here, $\gamma_{\mathbf{z}}^m$ represents the median of $\{\|\mathbf{z}_i - \mathbf{z}_j\|\}_{1 \le i < j \le n}$, and $\mathbf{z}$ is either $\mathbf{x}$ or $\mathbf{y}$. We repeat the simulations in Examples 1 to 3 of the main paper.

For Example 1, we display the kernel density curves of the test statistics for the two scenarios in Figures S1-S6 for different choices of $c_0$, respectively. It is clear that the results are consistent across different values of $c_0$: the normal approximation performs poorly when the dimensions are small, while the asymptotic distribution progressively converges more accurately to normality as the dimensions increase.

For Examples 2 and 3, we report the empirical powers for Models (I) to (VI) in Table S1 and Table S2. We can see that, when there exists linear dependences in Models (I) and (IV), all the empirical powers can still approach one, which implies that all tests can easily detect linear dependence, even when the dimension significantly exceeds the sample size. However, when there only exist higher orders of dependences in the remaining models, we observe that the empirical powers decay as the dimensions increase. Moreover, there exists a faster decay of powers in Models (III) and (VI) compared with Models (II) and (V).

It is worth noting that when the dimensions are not too large, i.e., $p = 30$ in Model (II) and $d = 6$ in Model (IV), when the multiplier for bandwidth parameters, denoted as $c_0$, is set to either 1.5 or 2, the test using a Gaussian kernel exhibits a slight reduction in power. Consequently, as a practical recommendation, we propose opting for $c_0 = 1$, which aligns with the sample medians precisely.

---

[*]Corresponding author

37th Conference on Neural Information Processing Systems (NeurIPS 2023).

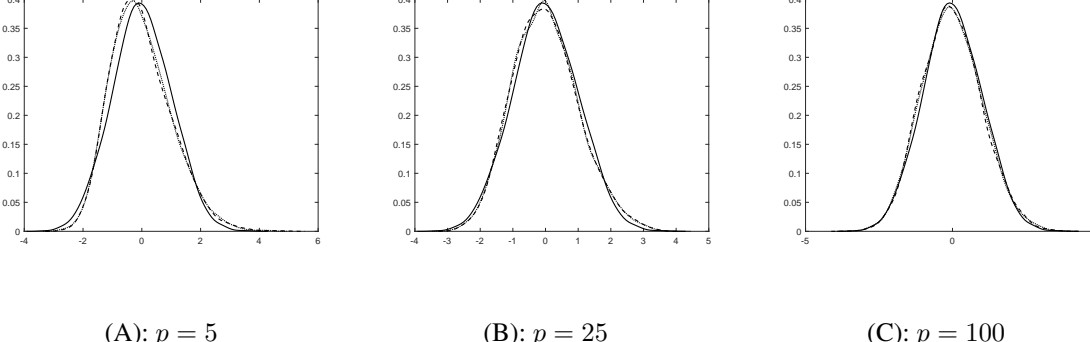

(A): $p = 5$                    (B): $p = 25$                    (C): $p = 100$

Figure S1: The kernel densities of the test statistics under the null hypothesis computed from 5000 simulations for Example 1. We fix $q = 1$ and vary $p$ from 5, 25, 100. The horizontal axes represent the observed values of the test statistics, and the vertical axes represent the kernel densities of those values. We choose two kinds of commonly used kernels to implement the tests, i.e., Gaussian (dashed line) and Laplacian (dotted line). The bandwidth parameters are set as $\gamma_{\mathbf{x}} = 0.5\gamma_{\mathbf{x}}^m$ and $\gamma_{\mathbf{y}} = 0.5\gamma_{\mathbf{y}}^m$. The solid line is the reference curve, which is the density of the standard normal distribution.

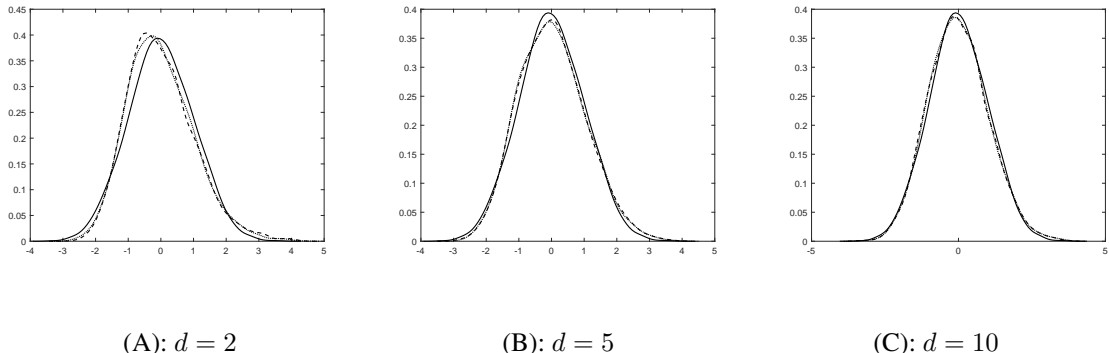

(A): $d = 2$                    (B): $d = 5$                    (C): $d = 10$

Figure S2: The kernel densities of the test statistics under the null hypothesis computed from 5000 simulations for Example 1. We set $p = q = d$ and vary $d$ from 2, 5, 10. The horizontal axes represent the observed values of the test statistics, and the vertical axes represent the kernel densities of those values. We choose two kinds of commonly used kernels to implement the tests, i.e., Gaussian (dashed line) and Laplacian (dotted line). The bandwidth parameters are set as $\gamma_{\mathbf{x}} = 0.5\gamma_{\mathbf{x}}^m$ and $\gamma_{\mathbf{y}} = 0.5\gamma_{\mathbf{y}}^m$. The solid line is the reference curve, which is the density of the standard normal distribution.

## S.2   Proof of Theorem 1

We first show that, under the null hypothesis, $\text{HSIC}_n(\mathbf{x}, \mathbf{y})$, which is defined in (2), has an equivalent form as $\text{HSIC}_n^*(\mathbf{x}, \mathbf{y})$, where $\text{HSIC}_n^*(\mathbf{x}, \mathbf{y})$ is defined as

$$
\begin{aligned}
\text{HSIC}_n^*(\mathbf{x}, \mathbf{y}) \;=\; & \frac{1}{n(n-1)} \sum_{(i_1, i_2)} H_{\mathbf{x}}(\mathbf{x}_{i_1}, \mathbf{x}_{i_2}) H_{\mathbf{y}}(\mathbf{y}_{i_1}, \mathbf{y}_{i_2}) \\
& - \frac{2}{n(n-1)(n-2)} \sum_{(i_1, i_2, i_3)} H_{\mathbf{x}}(\mathbf{x}_{i_1}, \mathbf{x}_{i_2}) H_{\mathbf{y}}(\mathbf{y}_{i_1}, \mathbf{y}_{i_3}) \\
& + \frac{1}{n(n-1)(n-2)(n-3)} \sum_{(i_1, i_2, i_3, i_4)} H_{\mathbf{x}}(\mathbf{x}_{i_1}, \mathbf{x}_{i_2}) H_{\mathbf{y}}(\mathbf{y}_{i_3}, \mathbf{y}_{i_4}). \quad \text{(S.1)}
\end{aligned}
$$

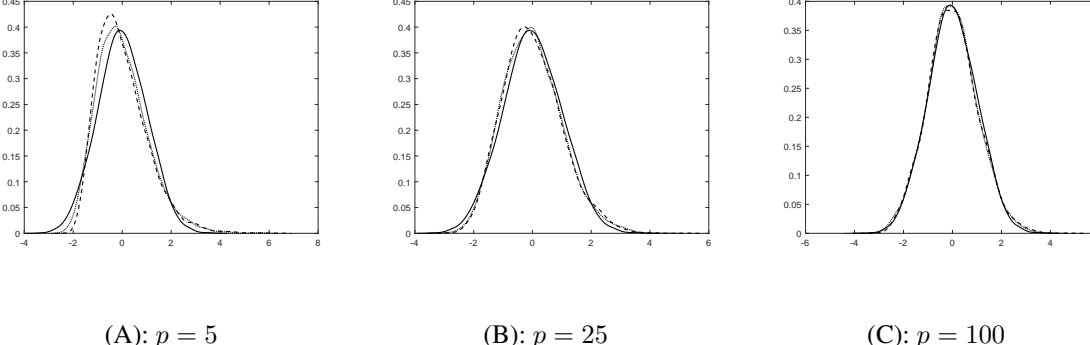

| (A): $p = 5$ | (B): $p = 25$ | (C): $p = 100$ |

Figure S3: The kernel densities of the test statistics under the null hypothesis computed from 5000 simulations for Example 1. We fix $q = 1$ and vary $p$ from 5, 25, 100. The horizontal axes represent the observed values of the test statistics, and the vertical axes represent the kernel densities of those values. We choose two kinds of commonly used kernels to implement the tests, i.e., Gaussian (dashed line) and Laplacian (dotted line). The bandwidth parameters are set as $\gamma_{\mathbf{x}} = 1.5\gamma_{\mathbf{x}}^m$ and $\gamma_{\mathbf{y}} = 1.5\gamma_{\mathbf{y}}^m$. The solid line is the reference curve, which is the density of the standard normal distribution.

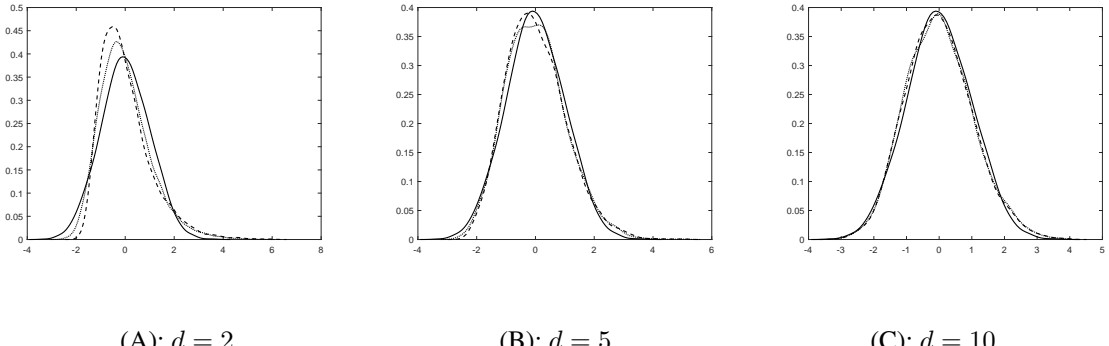

| (A): $d = 2$ | (B): $d = 5$ | (C): $d = 10$ |

Figure S4: The kernel densities of the test statistics under the null hypothesis computed from 5000 simulations for Example 1. We set $p = q = d$ and vary $d$ from 2, 5, 10. The horizontal axes represent the observed values of the test statistics, and the vertical axes represent the kernel densities of those values. We choose two kinds of commonly used kernels to implement the tests, i.e., Gaussian (dashed line) and Laplacian (dotted line). The bandwidth parameters are set as $\gamma_{\mathbf{x}} = 1.5\gamma_{\mathbf{x}}^m$ and $\gamma_{\mathbf{y}} = 1.5\gamma_{\mathbf{y}}^m$. The solid line is the reference curve, which is the density of the standard normal distribution.

Note that $H_{\mathbf{x}}(\mathbf{x}_1, \mathbf{x}_2)$ and $H_{\mathbf{y}}(\mathbf{y}_1, \mathbf{y}_2)$ are defined in (3). To verify this, we denote by $K_{i_1} = E\{K(\mathbf{x}_{i_1}, \mathbf{x}_{i_2}) \mid \mathbf{x}_{i_1}\}$ and $L_{i_1} = E\{L(\mathbf{y}_{i_1}, \mathbf{y}_{i_2}) \mid \mathbf{y}_{i_1}\}$. Then, by the symmetry of the kernels, i.e., $L(\mathbf{y}_1, \mathbf{y}_2) = L(\mathbf{y}_2, \mathbf{y}_1)$, we have $H_{\mathbf{y}}(\mathbf{y}_{i_1}, \mathbf{y}_{i_2}) = L(\mathbf{y}_{i_1}, \mathbf{y}_{i_2}) - L_{i_1} - L_{i_2} + EL(\mathbf{y}_{i_1}, \mathbf{y}_{i_2})$. In addition, we have

$$\sum_{(i_1, i_2)} H_{\mathbf{x}}(\mathbf{x}_{i_1}, \mathbf{x}_{i_2})(L_{i_1} + L_{i_2}) = \frac{2}{n-2} \sum_{(i_1, i_2, i_3)} H_{\mathbf{x}}(\mathbf{x}_{i_1}, \mathbf{x}_{i_2})L_{i_1},$$

$$\frac{1}{n-3} \sum_{(i_1, i_2, i_3, i_4)} H_{\mathbf{x}}(\mathbf{x}_{i_1}, \mathbf{x}_{i_2})(L_{i_3} + L_{i_4}) = 2 \sum_{(i_1, i_2, i_3)} H_{\mathbf{x}}(\mathbf{x}_{i_1}, \mathbf{x}_{i_2})L_{i_3}.$$

Substituting these two equations into (S.1), we have

$$\text{HSIC}_n^*(\mathbf{x}, \mathbf{y}) = \frac{1}{n(n-1)} \sum_{(i_1, i_2)} H_{\mathbf{x}}(\mathbf{x}_{i_1}, \mathbf{x}_{i_2})L(\mathbf{y}_{i_1}, \mathbf{y}_{i_2})$$

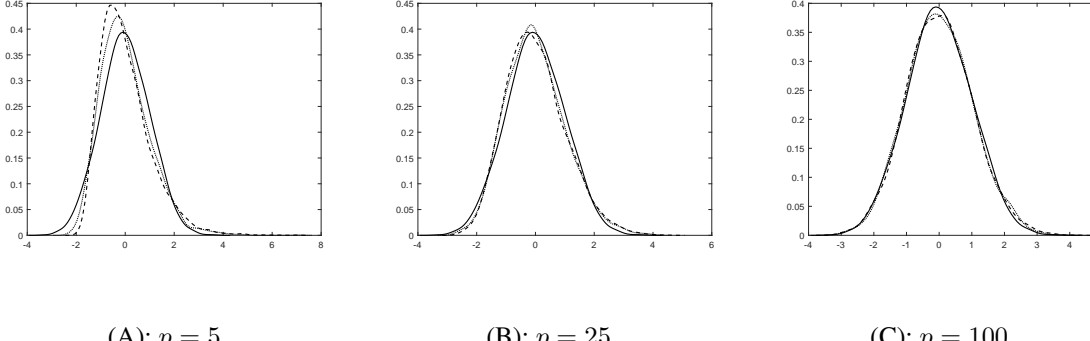

| (A): $p = 5$ | (B): $p = 25$ | (C): $p = 100$ |

Figure S5: The kernel densities of the test statistics under the null hypothesis computed from 5000 simulations for Example 1. We fix $q = 1$ and vary $p$ from 5, 25, 100. The horizontal axes represent the observed values of the test statistics, and the vertical axes represent the kernel densities of those values. We choose two kinds of commonly used kernels to implement the tests, i.e., Gaussian (dashed line) and Laplacian (dotted line). The bandwidth parameters are set as $\gamma_{\mathbf{x}} = 2\gamma_{\mathbf{x}}^m$ and $\gamma_{\mathbf{y}} = 2\gamma_{\mathbf{y}}^m$. The solid line is the reference curve, which is the density of the standard normal distribution.

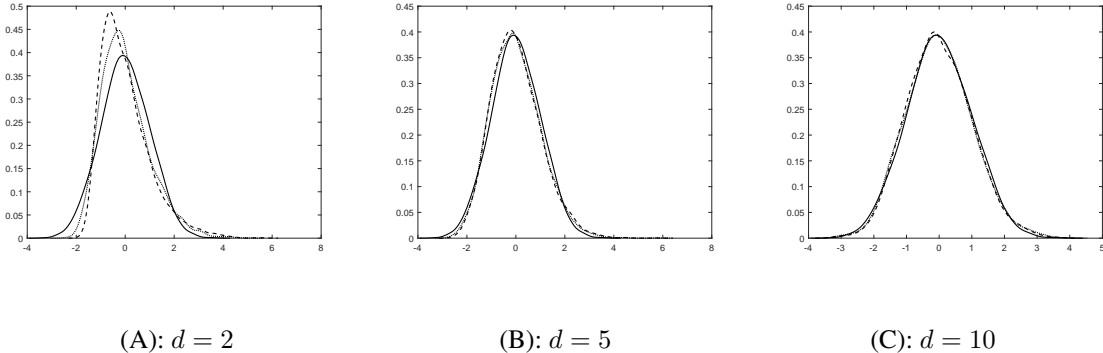

| (A): $d = 2$ | (B): $d = 5$ | (C): $d = 10$ |

Figure S6: The kernel densities of the test statistics under the null hypothesis computed from 5000 simulations for Example 1. We set $p = q = d$ and vary $d$ from 2, 5, 10. The horizontal axes represent the observed values of the test statistics, and the vertical axes represent the kernel densities of those values. We choose two kinds of commonly used kernels to implement the tests, i.e., Gaussian (dashed line) and Laplacian (dotted line). The bandwidth parameters are set as $\gamma_{\mathbf{x}} = 2\gamma_{\mathbf{x}}^m$ and $\gamma_{\mathbf{y}} = 2\gamma_{\mathbf{y}}^m$. The solid line is the reference curve, which is the density of the standard normal distribution.

$$-\frac{2}{n(n-1)(n-2)} \sum_{(i_1, i_2, i_3)} H_{\mathbf{x}}(\mathbf{x}_{i_1}, \mathbf{x}_{i_2}) L(\mathbf{y}_{i_1}, \mathbf{y}_{i_3})$$

$$+\frac{1}{n(n-1)(n-2)(n-3)} \sum_{(i_1, i_2, i_3, i_4)} H_{\mathbf{x}}(\mathbf{x}_{i_1}, \mathbf{x}_{i_2}) L(\mathbf{y}_{i_3}, \mathbf{y}_{i_4}). \quad \text{(S.2)}$$

Now with the fact that $K(\mathbf{x}_1, \mathbf{x}_2) = K(\mathbf{x}_2, \mathbf{x}_1)$ and $H_{\mathbf{x}}(\mathbf{x}_{i_1}, \mathbf{x}_{i_2}) = K(\mathbf{x}_{i_1}, \mathbf{x}_{i_2}) - K_{i_1} - K_{i_2} + EK(\mathbf{x}_{i_1}, \mathbf{x}_{i_2})$, we have

$$\sum_{(i_1, i_2)} (K_{i_1} + K_{i_2}) L(\mathbf{y}_{i_1}, \mathbf{y}_{i_2}) = \frac{2}{n-2} \sum_{(i_1, i_2, i_3)} K_{i_1} L(\mathbf{y}_{i_1}, \mathbf{y}_{i_3}),$$

$$\frac{1}{n-3} \sum_{(i_1, i_2, i_3, i_4)} (K_{i_1} + K_{i_2}) L(\mathbf{y}_{i_3}, \mathbf{y}_{i_4}) = 2 \sum_{(i_1, i_2, i_3)} K_{i_2} L(\mathbf{y}_{i_1}, \mathbf{y}_{i_3}).$$

Table S1: The empirical powers of different tests for Models (I)-(III) in Example 2. The significance level is 0.05. We fix $q = 1$ and vary $p$ from 30, 50, 100, 200, 500, 1000. The bandwidth parameters are set as $\gamma_{\mathbf{x}} = c_0 \gamma_{\mathbf{x}}^m$, $\gamma_{\mathbf{y}} = c_0 \gamma_{\mathbf{y}}^m$ and we evaluate different $c_0 = 0.5, 1, 1.5, 2$.

| Model | $c_0$ | Test | $p$ | | | | | |
|---|---|---|---|---|---|---|---|---|
| | | | 30 | 50 | 100 | 200 | 500 | 1000 |
| (I) | 0.5 | Gaussian | 1.000 | 1.000 | 1.000 | 0.994 | 0.914 | 0.710 |
| | | Laplacian | 1.000 | 1.000 | 1.000 | 0.996 | 0.900 | 0.696 |
| | 1 | Gaussian | 1.000 | 1.000 | 1.000 | 1.000 | 0.998 | 0.954 |
| | | Laplacian | 1.000 | 1.000 | 1.000 | 1.000 | 0.994 | 0.916 |
| | 1.5 | Gaussian | 1.000 | 1.000 | 1.000 | 1.000 | 1.000 | 0.992 |
| | | Laplacian | 1.000 | 1.000 | 1.000 | 1.000 | 0.998 | 0.950 |
| | 2 | Gaussian | 1.000 | 1.000 | 1.000 | 1.000 | 1.000 | 0.994 |
| | | Laplacian | 1.000 | 1.000 | 1.000 | 1.000 | 1.000 | 0.970 |
| (II) | 0.5 | Gaussian | 1.000 | 1.000 | 0.980 | 0.870 | 0.532 | 0.354 |
| | | Laplacian | 1.000 | 0.986 | 0.886 | 0.670 | 0.358 | 0.234 |
| | 1 | Gaussian | 0.934 | 0.774 | 0.484 | 0.318 | 0.184 | 0.132 |
| | | Laplacian | 0.998 | 0.978 | 0.834 | 0.596 | 0.314 | 0.234 |
| | 1.5 | Gaussian | 0.542 | 0.394 | 0.230 | 0.188 | 0.102 | 0.102 |
| | | Laplacian | 0.998 | 0.976 | 0.802 | 0.570 | 0.300 | 0.210 |
| | 2 | Gaussian | 0.282 | 0.236 | 0.140 | 0.140 | 0.084 | 0.080 |
| | | Laplacian | 0.998 | 0.966 | 0.786 | 0.554 | 0.282 | 0.212 |
| (III) | 0.5 | Gaussian | 0.050 | 0.050 | 0.064 | 0.052 | 0.056 | 0.060 |
| | | Laplacian | 0.058 | 0.048 | 0.062 | 0.048 | 0.058 | 0.058 |
| | 1 | Gaussian | 0.044 | 0.050 | 0.060 | 0.046 | 0.060 | 0.068 |
| | | Laplacian | 0.050 | 0.046 | 0.054 | 0.048 | 0.054 | 0.060 |
| | 1.5 | Gaussian | 0.042 | 0.050 | 0.048 | 0.046 | 0.056 | 0.046 |
| | | Laplacian | 0.046 | 0.048 | 0.054 | 0.044 | 0.054 | 0.062 |
| | 2 | Gaussian | 0.054 | 0.052 | 0.062 | 0.046 | 0.060 | 0.048 |
| | | Laplacian | 0.042 | 0.046 | 0.058 | 0.040 | 0.062 | 0.060 |

Then we can derive that $\mathrm{HSIC}_n^*(\mathbf{x}, \mathbf{y}) = \mathrm{HSIC}_n(\mathbf{x}, \mathbf{y})$ by substituting these two equations into (S.2).

Now we show that the last two terms in (S.1) are asymptotically negligible compared with the first term. Specifically, we show that,

$$\mathrm{HSIC}_n(\mathbf{x}, \mathbf{y}) = \frac{1}{n(n-1)} \sum_{i \neq j} H_{\mathbf{x}}(\mathbf{x}_i, \mathbf{x}_j) H_{\mathbf{y}}(\mathbf{y}_i, \mathbf{y}_j) + o_p\{n^{-1} H_{\mathbf{x}}(\mathbf{x}_1, \mathbf{x}_2) H_{\mathbf{y}}(\mathbf{y}_1, \mathbf{y}_2)\}. \quad \text{(S.3)}$$

It suffices to show that the second and the last term in (S.1) are both second order degenerate. In fact, by the definition of $H_{\mathbf{x}}(\mathbf{x}_{i_1}, \mathbf{x}_{i_2})$ and $H_{\mathbf{y}}(\mathbf{y}_{i_1}, \mathbf{y}_{i_2})$, it is straightforward that,

$$E\{H_{\mathbf{x}}(\mathbf{x}_{i_1}, \mathbf{x}_{i_2}) \mid \mathbf{x}_{i_1}\} = 0, \quad E\{H_{\mathbf{y}}(\mathbf{y}_{i_1}, \mathbf{y}_{i_3}) \mid \mathbf{y}_{i_1}\} = 0.$$

Then, under the independence of $\mathbf{x}$ and $\mathbf{y}$,

$$E\{H_{\mathbf{x}}(\mathbf{x}_{i_1}, \mathbf{x}_{i_2}) H_{\mathbf{y}}(\mathbf{y}_{i_1}, \mathbf{y}_{i_3}) \mid \mathbf{x}_{i_1}, \mathbf{y}_{i_1}\} = 0,$$
$$E\{H_{\mathbf{x}}(\mathbf{x}_{i_1}, \mathbf{x}_{i_2}) H_{\mathbf{y}}(\mathbf{y}_{i_1}, \mathbf{y}_{i_3}) \mid \mathbf{x}_{i_2}, \mathbf{y}_{i_2}\} = 0,$$
$$E\{H_{\mathbf{x}}(\mathbf{x}_{i_1}, \mathbf{x}_{i_2}) H_{\mathbf{y}}(\mathbf{y}_{i_1}, \mathbf{y}_{i_3}) \mid \mathbf{x}_{i_3}, \mathbf{y}_{i_3}\} = 0.$$

This implies that the second term in (S.1) is degenerate. In addition, we have

$$E\{H_{\mathbf{x}}(\mathbf{x}_{i_1}, \mathbf{x}_{i_2}) H_{\mathbf{y}}(\mathbf{y}_{i_1}, \mathbf{y}_{i_3}) \mid \mathbf{x}_{i_1}, \mathbf{x}_{i_2}, \mathbf{y}_{i_1}, \mathbf{y}_{i_2}\} = 0,$$
$$E\{H_{\mathbf{x}}(\mathbf{x}_{i_1}, \mathbf{x}_{i_2}) H_{\mathbf{y}}(\mathbf{y}_{i_1}, \mathbf{y}_{i_3}) \mid \mathbf{x}_{i_2}, \mathbf{x}_{i_3}, \mathbf{y}_{i_2}, \mathbf{y}_{i_3}\} = 0,$$
$$E\{H_{\mathbf{x}}(\mathbf{x}_{i_1}, \mathbf{x}_{i_2}) H_{\mathbf{y}}(\mathbf{y}_{i_1}, \mathbf{y}_{i_3}) \mid \mathbf{x}_{i_1}, \mathbf{x}_{i_3}, \mathbf{y}_{i_1}, \mathbf{y}_{i_3}\} = 0.$$

Table S2: The empirical powers of different tests for Models (IV)-(VI) in Example 3. The significance level is 0.05. We set $p = q = d$ and vary $d$ from 6, 10, 20, 50, 100, 200. The bandwidth parameters are set as $\gamma_{\mathbf{x}} = c_0 \gamma_{\mathbf{x}}^m$, $\gamma_{\mathbf{y}} = c_0 \gamma_{\mathbf{y}}^m$ and we evaluate different $c_0 = 0.5, 1, 1.5, 2$.

| Model | $c_0$ | Test | $d$ | | | | | |
|-------|-------|------|-----|-----|-----|-----|-----|-----|
| | | | 6 | 10 | 20 | 50 | 100 | 200 |
| (IV) | 0.5 | Gaussian | 1.000 | 1.000 | 1.000 | 1.000 | 1.000 | 1.000 |
| | | Laplacian | 1.000 | 1.000 | 1.000 | 1.000 | 1.000 | 1.000 |
| | 1 | Gaussian | 1.000 | 1.000 | 1.000 | 1.000 | 1.000 | 1.000 |
| | | Laplacian | 1.000 | 1.000 | 1.000 | 1.000 | 1.000 | 1.000 |
| | 1.5 | Gaussian | 1.000 | 1.000 | 1.000 | 1.000 | 1.000 | 1.000 |
| | | Laplacian | 1.000 | 1.000 | 1.000 | 1.000 | 1.000 | 1.000 |
| | 2 | Gaussian | 1.000 | 1.000 | 1.000 | 1.000 | 1.000 | 1.000 |
| | | Laplacian | 1.000 | 1.000 | 1.000 | 1.000 | 1.000 | 1.000 |
| (V) | 0.5 | Gaussian | 1.000 | 1.000 | 1.000 | 1.000 | 0.962 | 0.736 |
| | | Laplacian | 1.000 | 1.000 | 1.000 | 0.998 | 0.852 | 0.488 |
| | 1 | Gaussian | 1.000 | 1.000 | 0.944 | 0.440 | 0.242 | 0.140 |
| | | Laplacian | 1.000 | 1.000 | 1.000 | 0.904 | 0.578 | 0.282 |
| | 1.5 | Gaussian | 0.972 | 0.812 | 0.496 | 0.210 | 0.136 | 0.096 |
| | | Laplacian | 1.000 | 1.000 | 0.998 | 0.760 | 0.448 | 0.216 |
| | 2 | Gaussian | 0.712 | 0.496 | 0.310 | 0.148 | 0.104 | 0.088 |
| | | Laplacian | 1.000 | 1.000 | 0.992 | 0.674 | 0.390 | 0.192 |
| (VI) | 0.5 | Gaussian | 0.048 | 0.068 | 0.058 | 0.054 | 0.054 | 0.046 |
| | | Laplacian | 0.056 | 0.064 | 0.054 | 0.054 | 0.056 | 0.044 |
| | 1 | Gaussian | 0.072 | 0.048 | 0.048 | 0.06 | 0.062 | 0.042 |
| | | Laplacian | 0.062 | 0.066 | 0.046 | 0.06 | 0.054 | 0.042 |
| | 1.5 | Gaussian | 0.070 | 0.056 | 0.044 | 0.062 | 0.060 | 0.042 |
| | | Laplacian | 0.058 | 0.062 | 0.046 | 0.058 | 0.058 | 0.042 |
| | 2 | Gaussian | 0.064 | 0.054 | 0.046 | 0.062 | 0.062 | 0.044 |
| | | Laplacian | 0.058 | 0.060 | 0.048 | 0.056 | 0.058 | 0.042 |

That is, the second term in (S.1) is second order degenerate. Moreover, when $\mathbf{x}$ is independent of $\mathbf{y}$,

$$E\{H_{\mathbf{x}}(\mathbf{x}_1, \mathbf{x}_2) H_{\mathbf{y}}(\mathbf{y}_1, \mathbf{y}_3)\}^2 = E\{H_{\mathbf{x}}(\mathbf{x}_1, \mathbf{x}_2) H_{\mathbf{y}}(\mathbf{y}_1, \mathbf{y}_2)\}^2.$$

Then the second term in (S.1) is of order $o_p\{n^{-1} H_{\mathbf{x}}(\mathbf{x}_1, \mathbf{x}_2) H_{\mathbf{y}}(\mathbf{y}_1, \mathbf{y}_2)\}$. Similarly, we can also derive that the third term in (S.1) is also of order $o_p\{n^{-1} H_{\mathbf{x}}(\mathbf{x}_1, \mathbf{x}_2) H_{\mathbf{y}}(\mathbf{y}_1, \mathbf{y}_2)\}$. Therefore, (S.3) holds true, and $\text{HSIC}_n(\mathbf{x}, \mathbf{y})$ is asymptotically equal to a second order $U$-statistic.

In what follows, we show that, the standardized version of $\text{HSIC}_n(\mathbf{x}, \mathbf{y})$ converges in distribution to a standard normal distribution. First of all, we introduce several additional notations. Let $H(\mathbf{x}_1, \mathbf{y}_1, \mathbf{x}_2, \mathbf{y}_2)$ be the product of $H_{\mathbf{x}}(\mathbf{x}_1, \mathbf{x}_2)$ and $H_{\mathbf{x}}(\mathbf{x}_1, \mathbf{y}_2)$, and $G(\mathbf{x}_1, \mathbf{y}_1, \mathbf{x}_2, \mathbf{y}_2)$ be defined as

$$G(\mathbf{x}_1, \mathbf{y}_1, \mathbf{x}_2, \mathbf{y}_2) = E\{H(\mathbf{x}_3, \mathbf{y}_3, \mathbf{x}_1, \mathbf{y}_1) H(\mathbf{x}_3, \mathbf{y}_3, \mathbf{x}_2, \mathbf{y}_2) \mid \mathbf{x}_1, \mathbf{y}_1, \mathbf{x}_2, \mathbf{y}_2\}.$$

We further define $U_n$ as

$$U_n = \frac{1}{n(n-1)} \sum_{i \neq j} H_{\mathbf{x}}(\mathbf{x}_i, \mathbf{x}_j) H_{\mathbf{y}}(\mathbf{y}_i, \mathbf{y}_j).$$

Then $U_n$ is a degenerate $U$-statistic because

$$E\{H_{\mathbf{x}}(\mathbf{x}_{i_1}, \mathbf{x}_{i_2}) H_{\mathbf{y}}(\mathbf{y}_{i_1}, \mathbf{y}_{i_2}) \mid \mathbf{x}_{i_1}, \mathbf{y}_{i_1}\} = 0.$$

Then we have $U_n$ is of order $O_p\{n^{-1}H_{\mathbf{x}}(\mathbf{x}_1,\mathbf{x}_2)H_{\mathbf{y}}(\mathbf{y}_1,\mathbf{y}_2)\}$. We substitute $U_n$ into (S.3), we obtain that $\mathrm{HSIC}_n(\mathbf{x},\mathbf{y}) = U_n + o_p(U_n)$. Therefore, it suffices to study the asymptotic property for $U_n$.

According to Lemma 3.2 of Zheng (1996), if $E\{H^2(\mathbf{x}_1,\mathbf{y}_1,\mathbf{x}_2,\mathbf{y}_2)\} < \infty$, and

$$\frac{E\{G^2(\mathbf{x}_1,\mathbf{y}_1,\mathbf{x}_2,\mathbf{y}_2)\} + n^{-1}E\{H^4(\mathbf{x}_1,\mathbf{y}_1,\mathbf{x}_2,\mathbf{y}_2)\}}{E^2\{H^2(\mathbf{x}_1,\mathbf{y}_1,\mathbf{x}_2,\mathbf{y}_2)\}} \to 0, \tag{S.4}$$

$nU_n/[2E\{H^2(\mathbf{x}_1,\mathbf{y}_1,\mathbf{x}_2,\mathbf{y}_2)\}]^{1/2}$ is asymptotically standard normal. Then with the Slutsky's Theorem, $n\,\mathrm{HSIC}_n(\mathbf{x},\mathbf{y})/[2E\{H^2(\mathbf{x}_1,\mathbf{y}_1,\mathbf{x}_2,\mathbf{y}_2)\}]^{1/2}$ also has a limiting standard normal distribution. When $\mathbf{x}$ and $\mathbf{y}$ are independent, it follows that

$$E\{H^2(\mathbf{x}_1,\mathbf{y}_1,\mathbf{x}_2,\mathbf{y}_2)\} = E\{H_{\mathbf{x}}^2(\mathbf{x}_1,\mathbf{x}_2)H_{\mathbf{y}}^2(\mathbf{y}_1,\mathbf{y}_2)\} = E\{H_{\mathbf{x}}^2(\mathbf{x}_1,\mathbf{x}_2)\}E\{H_{\mathbf{y}}^2(\mathbf{y}_1,\mathbf{y}_2)\}.$$

According to Lyons (2013), we have $\mathrm{HSIC}(\mathbf{x},\mathbf{x}) = E\{H_{\mathbf{x}}^2(\mathbf{x}_1,\mathbf{x}_2)\}$ and $\mathrm{HSIC}(\mathbf{y},\mathbf{y}) = E\{H_{\mathbf{y}}^2(\mathbf{y}_1,\mathbf{y}_2)\}$. In addition, by Lemma 1 of Gao et al. (2021), $\mathrm{HSIC}_n(\mathbf{x},\mathbf{x})/\mathrm{HSIC}(\mathbf{x},\mathbf{x})$ and $\mathrm{HSIC}_n(\mathbf{y},\mathbf{y})/\mathrm{HSIC}(\mathbf{y},\mathbf{y})$ both converge in probability to one under assumption (4). Then we conclude that $2^{-1/2}n\,\mathrm{hCorr}_n^2(\mathbf{x},\mathbf{y})$ converges in distribution to a standard normal distribution with the Slutsky's Theorem.

Now it remains to verify (S.4). Under the null hypothesis, we have

$$\begin{aligned}
G(\mathbf{x}_1,\mathbf{y}_1,\mathbf{x}_2,\mathbf{y}_2) &= E\{H(\mathbf{x}_3,\mathbf{y}_3,\mathbf{x}_1,\mathbf{y}_1)H(\mathbf{x}_3,\mathbf{y}_3,\mathbf{x}_2,\mathbf{y}_2) \mid \mathbf{x}_1,\mathbf{y}_1,\mathbf{x}_2,\mathbf{y}_2\} \\
&= E\{H_{\mathbf{x}}(\mathbf{x}_1,\mathbf{x}_3)H_{\mathbf{x}}(\mathbf{x}_2,\mathbf{x}_3) \mid \mathbf{x}_1,\mathbf{x}_2\}E\{H_{\mathbf{y}}(\mathbf{y}_1,\mathbf{y}_3)H_{\mathbf{y}}(\mathbf{y}_2,\mathbf{y}_3) \mid \mathbf{y}_1,\mathbf{y}_2\},
\end{aligned}$$

which equals the product of $G_{\mathbf{x}}(\mathbf{x}_1,\mathbf{x}_2)$ and $G_{\mathbf{y}}(\mathbf{y}_1,\mathbf{y}_2)$. Recall that we have shown

$$E\{H^2(\mathbf{x}_1,\mathbf{y}_1,\mathbf{x}_2,\mathbf{y}_2)\} = E\{H_{\mathbf{x}}^2(\mathbf{x}_1,\mathbf{x}_2)\}E\{H_{\mathbf{y}}^2(\mathbf{y}_1,\mathbf{y}_2)\} = \mathrm{HSIC}(\mathbf{x},\mathbf{x})\mathrm{HSIC}(\mathbf{y},\mathbf{y}).$$

Then we have

$$\frac{E\{G^2(\mathbf{x}_1,\mathbf{y}_1,\mathbf{x}_2,\mathbf{y}_2)\}}{E^2\{H^2(\mathbf{x}_1,\mathbf{y}_1,\mathbf{x}_2,\mathbf{y}_2)\}} = \frac{E\{G_{\mathbf{x}}^2(\mathbf{x}_1,\mathbf{x}_2)\}E\{G_{\mathbf{y}}^2(\mathbf{y}_1,\mathbf{y}_2)\}}{\{\mathrm{HSIC}(\mathbf{x},\mathbf{x})\mathrm{HSIC}(\mathbf{y},\mathbf{y})\}^2},$$

which tends to 0 by assumption (4). In addition,

$$\frac{n^{-1}E\{H^4(\mathbf{x}_1,\mathbf{y}_1,\mathbf{x}_2,\mathbf{y}_2)\}}{E^2\{H^2(\mathbf{x}_1,\mathbf{y}_1,\mathbf{x}_2,\mathbf{y}_2)\}} = \frac{E\{H_{\mathbf{x}}^4(\mathbf{x}_1,\mathbf{x}_2)\}E\{H_{\mathbf{y}}^4(\mathbf{y}_1,\mathbf{y}_2)\}}{n\{\mathrm{HSIC}(\mathbf{x},\mathbf{x})\mathrm{HSIC}(\mathbf{y},\mathbf{y})\}^2} \to 0.$$

Therefore, we have verified (S.4), and the proof is completed. $\square$

## S.3 Proof of Theorem 2

Recall that the squared Hilbert-Schmidt correlation $\mathrm{hCorr}^2(\mathbf{x},\mathbf{y})$ is estimated as

$$\mathrm{hCorr}_n^2(\mathbf{x},\mathbf{y}) \overset{\mathrm{def}}{=} \frac{\mathrm{HSIC}_n(\mathbf{x},\mathbf{y})}{\sqrt{\mathrm{HSIC}_n(\mathbf{x},\mathbf{x})\mathrm{HSIC}_n(\mathbf{y},\mathbf{y})}}.$$

We deal with the denominator and the numerator, respectively. For the denominator, we show that $\mathrm{HSIC}(\mathbf{z},\mathbf{z}) \asymp d^{-\kappa_{\mathbf{z}}}$ and the ratio $\mathrm{HSIC}_n(\mathbf{z},\mathbf{z})/\mathrm{HSIC}(\mathbf{z},\mathbf{z})$ converges in probability to 1. By assumption (A1), we have

$$E(\|\mathbf{z}_1^* - \mathbf{z}_2^*\|^2 - E\|\mathbf{z}_1^* - \mathbf{z}_2^*\|^2)^{2k} \asymp d^{-k\kappa_{\mathbf{z}}}.$$

Then with Taylor's expansion and assumption (A2), we have

$$k_0(\|\mathbf{z}_1^* - \mathbf{z}_2^*\|^2) = k_0(E\|\mathbf{z}_1^* - \mathbf{z}_2^*\|^2) + k_0'(E\|\mathbf{z}_1^* - \mathbf{z}_2^*\|^2)(\|\mathbf{z}_1^* - \mathbf{z}_2^*\|^2 - E\|\mathbf{z}_1^* - \mathbf{z}_2^*\|^2) + O_p(d^{-\kappa_{\mathbf{z}}}).$$

Recall the definition of $H_{\mathbf{z}}(\mathbf{z}_1,\mathbf{z}_2)$, we derive that

$$\begin{aligned}
H_{\mathbf{z}}(\mathbf{z}_1,\mathbf{z}_2) &= k_0'(E\|\mathbf{z}_1^* - \mathbf{z}_2^*\|^2)\big\{\|\mathbf{z}_1^* - \mathbf{z}_2^*\|^2 - E(\|\mathbf{z}_1^* - \mathbf{z}_2^*\|^2 \mid \mathbf{z}_1^*) \\
&\qquad -E(\|\mathbf{z}_1^* - \mathbf{z}_2^*\|^2 \mid \mathbf{z}_2^*) + E(\|\mathbf{z}_1^* - \mathbf{z}_2^*\|^2)\big\} + O_p(d^{-\kappa_{\mathbf{z}}}) \\
&= 2k_0'(E\|\mathbf{z}_1^* - \mathbf{z}_2^*\|^2)(\mathbf{z}_1^* - E\mathbf{z}^*)^{\mathrm{T}}(\mathbf{z}_2^* - E\mathbf{z}^*) + O_p(d^{-\kappa_{\mathbf{z}}}).
\end{aligned}$$

Because $k_0'(E\|\mathbf{z}_1^* - \mathbf{z}_2^*\|^2)$ is bounded away from 0 to infinity, and $E\mathbf{z}^* = \mathbf{0}$, by assumption (A1), we have

$$E\{H_{\mathbf{z}}^{2k}(\mathbf{z}_1, \mathbf{z}_2)\} \asymp d^{-k\kappa_{\mathbf{z}}}.$$

This implies that $\mathrm{HSIC}(\mathbf{z}, \mathbf{z}) = E\{H_{\mathbf{z}}^2(\mathbf{z}_1, \mathbf{z}_2)\} \asymp d^{-\kappa_{\mathbf{z}}}$ and $E\{H_{\mathbf{z}}^4(\mathbf{z}_1, \mathbf{z}_2)\} \asymp d^{-2\kappa_{\mathbf{z}}}$. In addition, we conclude that

$$\frac{E\{H_{\mathbf{x}}^4(\mathbf{x}_1, \mathbf{x}_2)\}E\{H_{\mathbf{y}}^4(\mathbf{y}_1, \mathbf{y}_2)\}}{n\{\mathrm{HSIC}(\mathbf{x}, \mathbf{x})\mathrm{HSIC}(\mathbf{y}, \mathbf{y})\}^2} \to 0.$$

This guarantees the ratio consistency according to Lemma 1 of Gao et al. (2021). That is, $\mathrm{HSIC}_n(\mathbf{z}, \mathbf{z})/\mathrm{HSIC}(\mathbf{z}, \mathbf{z})$ converges in probability to 1. Therefore, to show $n\,\mathrm{hCorr}_n^2(\mathbf{x}, \mathbf{y}) \to \infty$ in probability, it suffices to show that $np^{\kappa_{\mathbf{x}}/2}q^{\kappa_{\mathbf{y}}/2}\mathrm{HSIC}_n(\mathbf{x}, \mathbf{y}) \to \infty$.

In what follows, we study the numerator of $\mathrm{hCorr}_n^2(\mathbf{x}, \mathbf{y})$, i.e., $\mathrm{HSIC}_n(\mathbf{x}, \mathbf{y})$. We show that, $np^{\kappa_{\mathbf{x}}/2}q^{\kappa_{\mathbf{y}}/2}\mathrm{HSIC}_n(\mathbf{x}, \mathbf{y}) \to \infty$ under the assumption of $n^{1/2}p^{\kappa_{\mathbf{x}}/2}q^{\kappa_{\mathbf{y}}/2}\mathrm{HSIC}(\mathbf{x}, \mathbf{y}) \to \infty$.

Recall that we have shown in (S.1) in the proof of Theorem 1 that, $\mathrm{HSIC}_n(\mathbf{x}, \mathbf{y})$ has an alternative expression,

$$
\begin{aligned}
\mathrm{HSIC}_n(\mathbf{x}, \mathbf{y}) \;=\; & \frac{1}{n(n-1)} \sum_{(i_1, i_2)} H_{\mathbf{x}}(\mathbf{x}_{i_1}, \mathbf{x}_{i_2}) H_{\mathbf{y}}(\mathbf{y}_{i_1}, \mathbf{y}_{i_2}) \\
& -\frac{2}{n(n-1)(n-2)} \sum_{(i_1, i_2, i_3)} H_{\mathbf{x}}(\mathbf{x}_{i_1}, \mathbf{x}_{i_2}) H_{\mathbf{y}}(\mathbf{y}_{i_1}, \mathbf{y}_{i_3}) \\
& +\frac{1}{n(n-1)(n-2)(n-3)} \sum_{(i_1, i_2, i_3, i_4)} H_{\mathbf{x}}(\mathbf{x}_{i_1}, \mathbf{x}_{i_2}) H_{\mathbf{y}}(\mathbf{y}_{i_3}, \mathbf{y}_{i_4}).
\end{aligned}
$$

We define the three summations in the above display as $U_{n1}$, $U_{n2}$, and $U_{n3}$, respectively. We first show that $U_{n2}$ and $U_{n3}$ are both of order $O_p(n^{-1}p^{-\kappa_{\mathbf{x}}/2}q^{-\kappa_{\mathbf{y}}/2})$. In fact, because

$$
\begin{aligned}
E\{H_{\mathbf{x}}(\mathbf{x}_{i_1}, \mathbf{x}_{i_2}) H_{\mathbf{y}}(\mathbf{y}_{i_1}, \mathbf{y}_{i_3}) \mid \mathbf{x}_{i_1}, \mathbf{y}_{i_1}, \mathbf{x}_{i_2}\} &= H_{\mathbf{x}}(\mathbf{x}_{i_1}, \mathbf{x}_{i_2}) E\{H_{\mathbf{y}}(\mathbf{y}_{i_1}, \mathbf{y}_{i_3}) \mid \mathbf{y}_{i_1}\} = 0, \\
E\{H_{\mathbf{x}}(\mathbf{x}_{i_1}, \mathbf{x}_{i_2}) H_{\mathbf{y}}(\mathbf{y}_{i_1}, \mathbf{y}_{i_3}) \mid \mathbf{x}_{i_2}, \mathbf{y}_{i_1}, \mathbf{y}_{i_3}\} &= H_{\mathbf{y}}(\mathbf{y}_{i_1}, \mathbf{y}_{i_3}) E\{H_{\mathbf{x}}(\mathbf{x}_{i_1}, \mathbf{x}_{i_2}) \mid \mathbf{x}_{i_2}\} = 0,
\end{aligned}
$$

we have

$$
\begin{aligned}
E\{H_{\mathbf{x}}(\mathbf{x}_{i_1}, \mathbf{x}_{i_2}) H_{\mathbf{y}}(\mathbf{y}_{i_1}, \mathbf{y}_{i_3}) \mid \mathbf{x}_{i_1}, \mathbf{y}_{i_1}\} &= 0, \\
E\{H_{\mathbf{x}}(\mathbf{x}_{i_1}, \mathbf{x}_{i_2}) H_{\mathbf{y}}(\mathbf{y}_{i_1}, \mathbf{y}_{i_3}) \mid \mathbf{x}_{i_2}\} &= 0, \\
E\{H_{\mathbf{x}}(\mathbf{x}_{i_1}, \mathbf{x}_{i_2}) H_{\mathbf{y}}(\mathbf{y}_{i_1}, \mathbf{y}_{i_3}) \mid \mathbf{y}_{i_3}\} &= 0.
\end{aligned}
$$

This implies that $U_{n2}$ is a degenerate $U$-statistic with mean 0. In addition, by the Cauchy-Schwarz inequality, we obtain that

$$
\begin{aligned}
\mathrm{var}\{H_{\mathbf{x}}(\mathbf{x}_{i_1}, \mathbf{x}_{i_2}) H_{\mathbf{y}}(\mathbf{y}_{i_1}, \mathbf{y}_{i_3})\} &= E\{H_{\mathbf{x}}^2(\mathbf{x}_{i_1}, \mathbf{x}_{i_2}) H_{\mathbf{y}}^2(\mathbf{y}_{i_1}, \mathbf{y}_{i_3})\} \\
&\leq E\{H_{\mathbf{x}}^2(\mathbf{x}_1, \mathbf{x}_2)\}E\{H_{\mathbf{y}}^2(\mathbf{y}_1, \mathbf{y}_2)\} + [\mathrm{var}\{H_{\mathbf{x}}^2(\mathbf{x}_1, \mathbf{x}_2)\}\mathrm{var}\{H_{\mathbf{y}}^2(\mathbf{y}_1, \mathbf{y}_2)\}]^{1/2}.
\end{aligned}
$$

We have shown that $E\{H_{\mathbf{z}}^{2k}(\mathbf{z}_1, \mathbf{z}_2)\} \asymp d^{-k\kappa_{\mathbf{z}}}$. Then we have

$$\mathrm{var}\{H_{\mathbf{z}}^2(\mathbf{z}_1, \mathbf{z}_2)\} \leq E\{H_{\mathbf{z}}^4(\mathbf{z}_1, \mathbf{z}_2)\} = O(d^{-2\kappa_{\mathbf{z}}}).$$

Therefore, the variance of $H_{\mathbf{x}}(\mathbf{x}_{i_1}, \mathbf{x}_{i_2}) H_{\mathbf{y}}(\mathbf{y}_{i_1}, \mathbf{y}_{i_3})$ is bounded by $O(p^{-\kappa_{\mathbf{x}}}q^{-\kappa_{\mathbf{y}}})$. Subsequently, we have $U_{n2}$ is of order $O_p(n^{-1}p^{-\kappa_{\mathbf{x}}/2}q^{-\kappa_{\mathbf{y}}/2})$.

Similarly, we can derive that $U_{n3}$ is also with mean 0 and of order $O_p(n^{-1}p^{-\kappa_{\mathbf{x}}/2}q^{-\kappa_{\mathbf{y}}/2})$. Now it remains to calculate the order of $U_{n1}$. Similar as bounding the variance of $H_{\mathbf{x}}(\mathbf{x}_{i_1}, \mathbf{x}_{i_2}) H_{\mathbf{y}}(\mathbf{y}_{i_1}, \mathbf{y}_{i_3})$, we also have, the variance of $H_{\mathbf{x}}(\mathbf{x}_{i_1}, \mathbf{x}_{i_2}) H_{\mathbf{y}}(\mathbf{y}_{i_1}, \mathbf{y}_{i_2})$ is bounded by $O(p^{-\kappa_{\mathbf{x}}}q^{-\kappa_{\mathbf{y}}})$. Then under the alternative hypothesis,

$$U_{n1} = E\{H_{\mathbf{x}}(\mathbf{x}_1, \mathbf{x}_2) H_{\mathbf{y}}(\mathbf{y}_1, \mathbf{y}_2)\} + O_p(n^{-1/2}p^{-\kappa_{\mathbf{x}}/2}q^{-\kappa_{\mathbf{y}}/2}).$$

Combing with the orders of $U_{n2}$ and $U_{n3}$, we obtain that

$$\mathrm{HSIC}_n(\mathbf{x}, \mathbf{y}) = \mathrm{HSIC}(\mathbf{x}, \mathbf{y}) + O_p(n^{-1/2}p^{-\kappa_{\mathbf{x}}/2}q^{-\kappa_{\mathbf{y}}/2}).$$

Therefore, when $n^{1/2}p^{\kappa_\mathbf{x}/2}q^{\kappa_\mathbf{y}/2}\mathrm{HSIC}(\mathbf{x},\mathbf{y}) \to \infty$,

$$\mathrm{HSIC}_n(\mathbf{x},\mathbf{y}) = \mathrm{HSIC}(\mathbf{x},\mathbf{y})\{1 + o_p(1)\}.$$

Subsequently, we have

$$np^{\kappa_\mathbf{x}/2}q^{\kappa_\mathbf{y}/2}\mathrm{HSIC}_n(\mathbf{x},\mathbf{y}) = np^{\kappa_\mathbf{x}/2}q^{\kappa_\mathbf{y}/2}\mathrm{HSIC}(\mathbf{x},\mathbf{y})\{1 + o_p(1)\},$$

which converges in probability to infinity. Recall that we have shown $\mathrm{HSIC}(\mathbf{z},\mathbf{z}) = E\{H_\mathbf{z}^2(\mathbf{z}_1,\mathbf{z}_2)\} \asymp d^{-\kappa_\mathbf{z}}$. Then we have

$$\mathrm{hCorr}^2(\mathbf{x},\mathbf{y}) = \frac{\mathrm{HSIC}(\mathbf{x},\mathbf{y})}{\sqrt{\mathrm{HSIC}(\mathbf{x},\mathbf{x})\mathrm{HSIC}(\mathbf{y},\mathbf{y})}} \asymp p^{\kappa_\mathbf{x}/2}q^{\kappa_\mathbf{y}/2}\mathrm{HSIC}(\mathbf{x},\mathbf{y}).$$

Therefore, $n^{1/2}p^{\kappa_\mathbf{x}/2}q^{\kappa_\mathbf{y}/2}\mathrm{HSIC}(\mathbf{x},\mathbf{y}) \to \infty$ is equivalent to $n^{1/2}\mathrm{hCorr}^2(\mathbf{x},\mathbf{y}) \to \infty$. This completes the proof of Theorem 2. $\square$

## S.4  Proof of Proposition 1

We have shown in the proof of Theorem 2 that

$$E\{H_\mathbf{z}^{2k}(\mathbf{z}_1,\mathbf{z}_2)\} \asymp d^{-k\kappa_\mathbf{z}}.$$

Then we conclude that $\mathrm{HSIC}(\mathbf{z},\mathbf{z}) = E\{H_\mathbf{z}^2(\mathbf{z}_1,\mathbf{z}_2)\} \asymp d^{-\kappa_\mathbf{z}}$, which completes the proof of Proposition 1. $\square$

## S.5  Proof of Lemma 1

Let $\widetilde{l}(\cdot) = l(\|\cdot\|/\gamma_\mathbf{y})$. Then the kernel $L(\mathbf{y}_1,\mathbf{y}_2)$ can be written as $\widetilde{l}(\mathbf{y}_1 - \mathbf{y}_2)$. By applying Theorem 9 of Sriperumbudur et al. (2010), we have $\widetilde{l}(\cdot)/\widetilde{l}(\mathbf{0})$ is the characteristic function of a random vector supported on $\mathbb{R}^q$. Then $L(\mathbf{y}_1,\mathbf{y}_2) = \widetilde{l}(\mathbf{y}_1 - \mathbf{y}_2)$ can be written as

$$L(\mathbf{y}_1,\mathbf{y}_2) = \widetilde{l}(\mathbf{0})\int \exp\{i(\mathbf{y}_1 - \mathbf{y}_2)^\mathrm{T}\mathbf{u}\}\omega(\mathbf{u})d\mathbf{u},$$

where $\omega(\mathbf{u})$ is the p.d.f. of a random vector supported on $\mathbb{R}^q$. Then we obtain that

$$
\begin{aligned}
\mathrm{MD}^2(\mathbf{x}\mid\mathbf{y}) &= E\{(\mathbf{x}_1 - E\mathbf{x})^\mathrm{T}(\mathbf{x}_2 - E\mathbf{x})L(\mathbf{y}_1,\mathbf{y}_2)\} \\
&= \widetilde{l}(\mathbf{0})E\left[(\mathbf{x}_1 - E\mathbf{x})^\mathrm{T}(\mathbf{x}_2 - E\mathbf{x})\int \exp\{i(\mathbf{y}_1 - \mathbf{y}_2)^\mathrm{T}\mathbf{u}\}\omega(\mathbf{u})d\mathbf{u}\right] \\
&= \widetilde{l}(\mathbf{0})\int \|E\{(\mathbf{x} - E\mathbf{x})\exp(i\mathbf{y}^\mathrm{T}\mathbf{u})\}\|^2\omega(\mathbf{u})d\mathbf{u}.
\end{aligned}
$$

Because $\omega(\mathbf{u})$ is the p.d.f. of a random vector supported on $\mathbb{R}^q$, we have $\omega(\mathbf{u}) > 0$ for all $\mathbf{u} \in \mathbb{R}^q$. Then we have $\mathrm{MD}^2(\mathbf{x}\mid\mathbf{y}) = 0$ if and only if

$$E\{(\mathbf{x} - E\mathbf{x})\exp(i\mathbf{y}^\mathrm{T}\mathbf{u})\} = 0$$

for all $\mathbf{u} \in \mathbb{R}^q$, which is equivalent to $E(\mathbf{x}\mid\mathbf{y}) = E\mathbf{x}$. This completes the proof. $\square$

## S.6  Proof of Theorem 3

First of all, because we have assumed that $E\mathbf{z}^* = \mathbf{0}$ without loss of generality, then

$$E(\|\mathbf{z}_1^* - \mathbf{z}_2^*\|^2 - E\|\mathbf{z}_1^* - \mathbf{z}_2^*\|^2)^{2k} = E\{\|\mathbf{z}_1^*\|^2 - E(\|\mathbf{z}^*\|^2) + \|\mathbf{z}_2^*\|^2 - E(\|\mathbf{z}^*\|^2) - 2\mathbf{z}_1^{*\mathrm{T}}\mathbf{z}_2^*\}^{2k}.$$

which is of order $O(d^{-k\kappa_\mathbf{z}})$ by Assumption (A1). Then we have, for each $s \in \mathbb{N}^+$,

$$(\|\mathbf{z}_1^* - \mathbf{z}_2^*\|^2 - E\|\mathbf{z}_1^* - \mathbf{z}_2^*\|^2)^s = O_p(d^{-s\kappa_\mathbf{z}/2}).$$

Then with Taylor's expansion, we have,

$$K(\mathbf{x}_1,\mathbf{x}_2) = \sum_{i=0}^{s}(i!)^{-1}k_0^{(i)}(\|\mathbf{x}_1^* - \mathbf{x}_2^*\|^2 - E\|\mathbf{x}_1^* - \mathbf{x}_2^*\|^2)^i + o_p(p^{-s\kappa_\mathbf{x}/2}), \tag{S.5}$$

where $k_0^{(i)}$ is the $i$-th derivative of $k_0(\cdot)$ evaluated at $E\|\mathbf{x}_1^* - \mathbf{x}_2^*\|^2$. By the multinomial theorem, with the fact that $E\mathbf{z}^* = \mathbf{0}$, we have

$$
(\|\mathbf{x}_1^* - \mathbf{x}_2^*\|^2 - E\|\mathbf{x}_1^* - \mathbf{x}_2^*\|^2)^i
$$
$$
= \frac{i!}{a!b!c!} \sum_{a+b+c=i} \{\|\mathbf{x}_1^*\|^2 - E(\|\mathbf{x}_1^*\|^2)\}^a \{\|\mathbf{x}_2^*\|^2 - E(\|\mathbf{x}_2^*\|^2)\}^b (-2\mathbf{x}_1^{*\mathrm{T}}\mathbf{x}_2^*)^c. \quad \text{(S.6)}
$$

By the definition of $\mathrm{HSIC}(\mathbf{x}, \mathbf{y})$, it can be verified that

$$
\mathrm{HSIC}(\mathbf{x}, \mathbf{y}) = \mathrm{cov}\{H_{\mathbf{x}}(\mathbf{x}_1, \mathbf{x}_2)H_{\mathbf{y}}(\mathbf{y}_1, \mathbf{y}_2)\} = E\{K(\mathbf{x}_1, \mathbf{x}_2)H_{\mathbf{y}}(\mathbf{y}_1, \mathbf{y}_2)\}.
$$

Combing this with (S.5) and (S.6), when $p \to \infty$ and $q$ is fixed, we obtain that

$$
\mathrm{HSIC}(\mathbf{x}, \mathbf{y}) = \sum_{i=0}^{s}(i!)^{-1}k_0^{(i)} \sum_{a+b+c=i} \frac{i!}{a!b!c!} E\left[ \{\|\mathbf{x}_1^*\|^2 - E(\|\mathbf{x}_1^*\|^2)\}^a \right.
$$
$$
\left. \{\|\mathbf{x}_2^*\|^2 - E(\|\mathbf{x}_2^*\|^2)\}^b (-2\mathbf{x}_1^{*\mathrm{T}}\mathbf{x}_2^*)^c H_{\mathbf{y}}(\mathbf{y}_1, \mathbf{y}_2) \right] + o_p(p^{-s\kappa_{\mathbf{x}}/2}). \quad \text{(S.7)}
$$

Because $E(\mathbf{x}^{\otimes t} \mid \mathbf{y}) = E(\mathbf{x}^{\otimes t})$ for all $t < s$, we have, as long as $2a + c < s$,

$$
E\left[ \{\|\mathbf{x}_1^*\|^2 - E(\|\mathbf{x}_1^*\|^2)\}^a \{\|\mathbf{x}_2^*\|^2 - E(\|\mathbf{x}_2^*\|^2)\}^b (-2\mathbf{x}_1^{*\mathrm{T}}\mathbf{x}_2^*)^c H_{\mathbf{y}}(\mathbf{y}_1, \mathbf{y}_2) \mid (\mathbf{y}_1, \mathbf{x}_2, \mathbf{y}_2) \right]
$$
$$
= \{\|\mathbf{x}_2^*\|^2 - E(\|\mathbf{x}_2^*\|^2)\}^b H_{\mathbf{y}}(\mathbf{y}_1, \mathbf{y}_2) E\left[ \{\|\mathbf{x}_1^*\|^2 - E(\|\mathbf{x}_1^*\|^2)\}^a \right] E\{(-2\mathbf{x}_1^{*\mathrm{T}}\mathbf{x}_2^*)^c \mid \mathbf{x}_2\}.
$$

Then with the law of iterated expectation, as long as $2a + c < s$, we have

$$
E\left[ \{\|\mathbf{x}_1^*\|^2 - E(\|\mathbf{x}_1^*\|^2)\}^a \{\|\mathbf{x}_2^*\|^2 - E(\|\mathbf{x}_2^*\|^2)\}^b (-2\mathbf{x}_1^{*\mathrm{T}}\mathbf{x}_2^*)^c H_{\mathbf{y}}(\mathbf{y}_1, \mathbf{y}_2) \right]
$$
$$
= E\left[ \{\|\mathbf{x}_1^*\|^2 - E(\|\mathbf{x}_1^*\|^2)\}^a \right] E\left[ \{\|\mathbf{x}_2^*\|^2 - E(\|\mathbf{x}_2^*\|^2)\}^b H_{\mathbf{y}}(\mathbf{y}_1, \mathbf{y}_2) E\{(-2\mathbf{x}_1^{*\mathrm{T}}\mathbf{x}_2^*)^c \mid \mathbf{x}_2\} \right].
$$

By using the law of iterated expectation again, we have

$$
E\left[ \{\|\mathbf{x}_2^*\|^2 - E(\|\mathbf{x}_2^*\|^2)\}^b H_{\mathbf{y}}(\mathbf{y}_1, \mathbf{y}_2) E\{(-2\mathbf{x}_1^{*\mathrm{T}}\mathbf{x}_2^*)^c \mid \mathbf{x}_2\} \right]
$$
$$
= E\left[ \{\|\mathbf{x}_2^*\|^2 - E(\|\mathbf{x}_2^*\|^2)\}^b E\{(-2\mathbf{x}_1^{*\mathrm{T}}\mathbf{x}_2^*)^c \mid \mathbf{x}_2\} E\{H_{\mathbf{y}}(\mathbf{y}_1, \mathbf{y}_2) \mid (\mathbf{x}_2, \mathbf{y}_2)\} \right],
$$

which equals zero because $E\{H_{\mathbf{y}}(\mathbf{y}_1, \mathbf{y}_2) \mid (\mathbf{x}_2, \mathbf{y}_2)\} = 0$. Therefore, we conclude that

$$
E\left[ \{\|\mathbf{x}_1^*\|^2 - E(\|\mathbf{x}_1^*\|^2)\}^a \{\|\mathbf{x}_2^*\|^2 - E(\|\mathbf{x}_2^*\|^2)\}^b (-2\mathbf{x}_1^{*\mathrm{T}}\mathbf{x}_2^*)^c H_{\mathbf{y}}(\mathbf{y}_1, \mathbf{y}_2) \right] = 0 \quad \text{(S.8)}
$$

for all $2a + c < s$. Similarly, (S.8) also holds true as long as $2b + c < s$. Therefore, we have, when $2\min(a, b) + c < s$, (S.8) holds true. It further implies that (S.8) holds true when $a + b + c < s$. This together with (S.7), imply that

$$
\mathrm{HSIC}(\mathbf{x}, \mathbf{y}) = k_0^{(s)} \sum_{2a+c=s} \frac{1}{a!a!c!} E\left[ \{\|\mathbf{x}_1^*\|^2 - E(\|\mathbf{x}_1^*\|^2)\}^a \right.
$$
$$
\left. \{\|\mathbf{x}_2^*\|^2 - E(\|\mathbf{x}_2^*\|^2)\}^a (-2\mathbf{x}_1^{*\mathrm{T}}\mathbf{x}_2^*)^c H_{\mathbf{y}}(\mathbf{y}_1, \mathbf{y}_2) \right] + o(p^{-s\kappa_{\mathbf{x}}/2}). \quad \text{(S.9)}
$$

In addition, $\mathrm{HSIC}(\mathbf{x}, \mathbf{y})$ is of order $O(p^{-s\kappa_{\mathbf{x}}/2})$. Moreover, similar to deriving (S.8), we can show that

$$
E\left\{ (\|\mathbf{x}_1^*\|^2)^{a_1}(\|\mathbf{x}_2^*\|^2)^{a_2}(-2\mathbf{x}_1^{*\mathrm{T}}\mathbf{x}_2^*)^c H_{\mathbf{y}}(\mathbf{y}_1, \mathbf{y}_2) \right\} = 0
$$

for all $a_1 + a_2 + c < s$. Then (S.9) can be further reduced to

$$
\mathrm{HSIC}(\mathbf{x}, \mathbf{y}) = k_0^{(s)} \sum_{2a+c=s} \frac{1}{a!a!c!} E\left\{ \|\mathbf{x}_1^*\|^{2a}\|\mathbf{x}_2^*\|^{2a}(-2\mathbf{x}_1^{*\mathrm{T}}\mathbf{x}_2^*)^c H_{\mathbf{y}}(\mathbf{y}_1, \mathbf{y}_2) \right\} + o(p^{-s\kappa_{\mathbf{x}}/2}).
$$

By the definition of $\mathrm{MD}^2(\mathbf{x} \mid \mathbf{y})$, we have

$$
\mathrm{MD}^2(\mathbf{x} \mid \mathbf{y}) = E\{(\mathbf{x}_1 - E\mathbf{x})^{\mathrm{T}}(\mathbf{x}_2 - E\mathbf{x})L(\mathbf{y}_1, \mathbf{y}_2)\} = E\{\mathbf{x}_1^{\mathrm{T}}\mathbf{x}_2 H_{\mathbf{y}}(\mathbf{y}_1, \mathbf{y}_2)\}.
$$

Then by noting that $(\mathbf{x}_1^{*\mathrm{T}}\mathbf{x}_2^*)^c = \{\mathbf{x}_1^{*\otimes c}\}^{\mathrm{T}}\mathbf{x}_2^{*\otimes c}$, we have

$$
\mathrm{HSIC}(\mathbf{x}, \mathbf{y}) = k_0^{(s)} \sum_{2a+c=s} \frac{(-2)^c}{a!a!c!} \mathrm{MD}^2(\mathbf{x}^{*\otimes c}\|\mathbf{x}^*\|^{2a} \mid \mathbf{y}) + o(p^{-s\kappa_{\mathbf{x}}/2}).
$$

This completes the proof of the first assertion.

When both dimensions $p$ and $q$ diverge to infinity, we first combine (S.5) and (S.6), as well as the fact that $(\mathbf{x}_1^{*\mathrm{T}}\mathbf{x}_2^*)^c = \{\mathbf{x}_1^{*\otimes c}\}^{\mathrm{T}}\mathbf{x}_2^{*\otimes c}$, to obtain that for each $s_1 \in \mathbb{N}^+$,

$$
\begin{aligned}
K(\mathbf{x}_1,\mathbf{x}_2) &= \sum_{i=0}^{s_1} \frac{k_0^{(i)}}{a_1!b_1!c_1!} \sum_{a_1+b_1+c_1=i} (-2)^{c_1} \Bigg[ \{\|\mathbf{x}_1^*\|^2 - E(\|\mathbf{x}_1^*\|^2)\}^{a_1}\{\mathbf{x}_1^{*\otimes c_1}\}^{\mathrm{T}} \\
&\qquad \{\|\mathbf{x}_2^*\|^2 - E(\|\mathbf{x}_2^*\|^2)\}^{b_1}\mathbf{x}_2^{*\otimes c_1} \Bigg] + o_p(p^{-s_1\kappa_{\mathbf{x}}/2}) \\
&= \sum_{i=0}^{s_1} \sum_{a_1+b_1+c_1=i} \frac{k_0^{(i)}(-2)^{c_1}}{a_1!b_1!c_1!} h(\mathbf{x}_1^*,a_1,c_1)^{\mathrm{T}}h(\mathbf{x}_2^*,b_1,c_1) + o_p(p^{-s_1\kappa_{\mathbf{x}}/2}),
\end{aligned}
$$

where $h(\mathbf{x}_1^*,a_1,c_1) = \{\|\mathbf{x}_1^*\|^2 - E(\|\mathbf{x}_1^*\|^2)\}^{a_1}\{\mathbf{x}_1^{*\otimes c_1}\}$. Then by the definition of $H_{\mathbf{x}}(\mathbf{x}_1,\mathbf{x}_2)$ in (3), we have

$$
\begin{aligned}
H_{\mathbf{x}}(\mathbf{x}_1,\mathbf{x}_2) &= \sum_{i=0}^{s_1} \sum_{a_1+b_1+c_1=i} \frac{k_0^{(i)}(-2)^{c_1}}{a_1!b_1!c_1!} \Bigg[ \{h(\mathbf{x}_1^*,a_1,c_1) - Eh(\mathbf{x}_1^*,a_1,c_1)\}^{\mathrm{T}} \\
&\qquad \{h(\mathbf{x}_2^*,b_1,c_1) - Eh(\mathbf{x}_2^*,b_1,c_1)\} \Bigg] + o_p(p^{-s_1\kappa_{\mathbf{x}}/2}).
\end{aligned}
$$

With similar arguments, we can also derive that, for each $s_2 \in \mathbb{N}^+$,

$$
\begin{aligned}
H_{\mathbf{y}}(\mathbf{y}_1,\mathbf{y}_2) &= \sum_{j=0}^{s_2} \sum_{a_2+b_2+c_2=j} \frac{l_0^{(j)}(-2)^{c_2}}{a_2!b_2!c_2!} \Bigg[ \{h(\mathbf{y}_1^*,a_2,c_2) - Eh(\mathbf{y}_1^*,a_2,c_2)\}^{\mathrm{T}} \\
&\qquad \{h(\mathbf{y}_2^*,b_2,c_2) - Eh(\mathbf{y}_2^*,b_2,c_2)\} \Bigg] + o_p(q^{-s_2\kappa_{\mathbf{y}}/2}),
\end{aligned}
$$

where $l_0^{(j)}$ is the $j$-th derivative of $l_0(\cdot)$ evaluated at $E\|\mathbf{y}_1^* - \mathbf{y}_2^*\|^2$. We now make an assertion that if $\mathrm{cov}(\mathbf{x}^{\otimes t_1}, \mathbf{y}^{\otimes t_2}) \neq \mathbf{0}$ only when $t_1 \geq s_1 \in \mathbb{N}^+$ and $t_2 \geq s_2 \in \mathbb{N}^+$, we have

$$
E\Bigg[ \{h(\mathbf{x}_1^*,a_1,c_1) - Eh(\mathbf{x}_1^*,a_1,c_1)\}^{\mathrm{T}}\{h(\mathbf{x}_2^*,b_1,c_1) - Eh(\mathbf{x}_2^*,b_1,c_1)\}
$$

$$
\{h(\mathbf{y}_1^*,a_2,c_2) - Eh(\mathbf{y}_1^*,a_2,c_2)\}^{\mathrm{T}}\{h(\mathbf{y}_2^*,b_2,c_2) - Eh(\mathbf{y}_2^*,b_2,c_2)\} \Bigg] = 0 \qquad (\mathrm{S}.10)
$$

as long as either $2\min(a_1,b_1) + c_1 < s_1$ or $2\min(a_2,b_2) + c_2 < s_2$ holds true. With (S.10), it is straightforward that $\mathrm{HSIC}(\mathbf{x},\mathbf{y}) = E\{H_{\mathbf{x}}(\mathbf{x}_1,\mathbf{x}_2)H_{\mathbf{y}}(\mathbf{y}_1,\mathbf{y}_2)\}$ equals

$$
\begin{aligned}
&\sum_{2a_1+c_1=s_1} \frac{k_0^{(s_1)}(-2)^{c_1}}{a_1!a_1!c_1!} \sum_{2a_2+c_2=s_2} \frac{l_0^{(s_2)}(-2)^{c_2}}{a_2!a_2!c_2!} E\Bigg[ \{h(\mathbf{x}_2^*,a_1,c_1) - Eh(\mathbf{x}_2^*,a_1,c_1)\}^{\mathrm{T}} \\
&\qquad \{h(\mathbf{x}_1^*,a_1,c_1) - Eh(\mathbf{x}_1^*,a_1,c_1)\}\{h(\mathbf{y}_1^*,a_2,c_2) - Eh(\mathbf{y}_1^*,a_2,c_2)\}^{\mathrm{T}} \\
&\qquad \{h(\mathbf{y}_2^*,a_2,c_2) - Eh(\mathbf{y}_2^*,a_2,c_2)\} \Bigg] + o(p^{-s_1\kappa_{\mathbf{x}}/2}q^{-s_2\kappa_{\mathbf{y}}/2}).
\end{aligned}
$$

In addition, the leading term is clearly of order $O(p^{-s_1\kappa_{\mathbf{x}}/2}q^{-s_2\kappa_{\mathbf{y}}/2})$. Let $(\mathbf{u}_1,\mathbf{v}_1)$ and $(\mathbf{u}_2,\mathbf{v}_2)$ be two independent copies of $(\mathbf{u},\mathbf{v})$ with arbitrary given dimensions. Because

$$
\begin{aligned}
&E\{(\mathbf{x}_2 - E\mathbf{x})^{\mathrm{T}}(\mathbf{x}_1 - E\mathbf{x})(\mathbf{y}_1 - E\mathbf{y})^{\mathrm{T}}(\mathbf{y}_2 - E\mathbf{y})\} \\
&= \mathrm{tr}[E\{(\mathbf{x}_1 - E\mathbf{x})(\mathbf{y}_1 - E\mathbf{y})^{\mathrm{T}}(\mathbf{y}_2 - E\mathbf{y})(\mathbf{x}_2 - E\mathbf{x})^{\mathrm{T}}\}],
\end{aligned}
$$

which equals $\|\mathrm{cov}(\mathbf{x},\mathbf{y}^{\mathrm{T}})\|_F^2$. Then we have, the leading term in $\mathrm{HSIC}(\mathbf{x},\mathbf{y})$ further equals

$$
\sum_{2a_1+c_1=s_1} \frac{k_0^{(s_1)}(-2)^{c_1}}{a_1!a_1!c_1!} \sum_{2a_2+c_2=s_2} \frac{l_0^{(s_2)}(-2)^{c_2}}{a_2!a_2!c_2!} \Big\|\mathrm{cov}\{h(\mathbf{x}^*,a_1,c_1), h(\mathbf{y}^*,a_2,c_2)^{\mathrm{T}}\}\Big\|_F^2.
$$

Recall that $h(\mathbf{x}^*, a_1, c_1) = \{\|\mathbf{x}^*\|^2 - E(\|\mathbf{x}^*\|^2)\}^{a_1}\{\mathbf{x}^{*\otimes c_1}\}$. It is clear that

$$\text{cov}\{h(\mathbf{x}^*, a_1, c_1), h(\mathbf{y}^*, a_2, c_2)^{\text{T}}\} = \text{cov}\{\|\mathbf{x}^*\|^{2a_1}\mathbf{x}^{*\otimes c_1}, \|\mathbf{y}^*\|^{2a_2}\mathbf{y}^{*\otimes c_2 \text{T}}\}$$

if $\text{cov}(\mathbf{x}^{\otimes t_1}, \mathbf{y}^{\otimes t_2}) \neq \mathbf{0}$ only when $t_1 \geq s_1 \in \mathbb{N}^+$ and $t_2 \geq s_2 \in \mathbb{N}^+$. Then we conclude that

$$
\begin{aligned}
\text{HSIC}(\mathbf{x}, \mathbf{y}) \quad = \quad & \sum_{2a_1+c_1=s_1} \sum_{2a_2+c_2=s_2} \frac{k_0^{(s_1)}(-2)^{c_1}}{a_1! a_1! c_1!} \frac{l_0^{(s_2)}(-2)^{c_2}}{a_2! a_2! c_2!} \\
& \left\| \text{cov}\{\|\mathbf{x}^*\|^{2a_1}\mathbf{x}^{*\otimes c_1}, \|\mathbf{y}^*\|^{2a_2}\mathbf{y}^{*\otimes c_2 \text{T}}\} \right\|_F^2 + o(p^{-s_1\kappa_{\mathbf{x}}/2}q^{-s_2\kappa_{\mathbf{y}}/2}).
\end{aligned}
$$

Now it remains to verify (S.10). Without loss of generality, we assume $a_1 \leq b_1$. Then when $2\min(a_1, b_1) + c_1 = 2a_1 + c_1 < s_1$,

$$E\left[\{h(\mathbf{x}_1^*, a_1, c_1) - Eh(\mathbf{x}_1^*, a_1, c_1)\}\{h(\mathbf{y}_1^*, a_2, c_2) - Eh(\mathbf{y}_1^*, a_2, c_2)\}^{\text{T}}\right] = 0.$$

With the law of iterated expectation, (S.10) holds true. Similarly, we have, (S.10) also holds true when $2\min(a_2, b_2) + c_2 < s_2$. This verifies that (S.10) holds true as long as either $2\min(a_1, b_1) + c_1 < s_1$ or $2\min(a_2, b_2) + c_2 < s_2$ holds true. Therefore, the proof of the second part is completed. $\square$