# OpenReview forum: "Statistical Insights into HSIC in High Dimensions"
_NeurIPS.cc/2023/Conference — NeurIPS 2023 poster_

### Official Review · Reviewer_TY3X · 2023-07-02

**Soundness:** 4 excellent
**Presentation:** 4 excellent
**Contribution:** 4 excellent
**Rating:** 8
**Confidence:** 4

**Summary:**

The paper investigate the performance of HSIC to test the independence of two random vectors. The focus is on high but not ultra high dimensional scenarios where the theory is lacking.  More specifically, the paper presents convergence rates for HSIC as the dimensions grow at different rates, and demonstrates how HSIC’s capacity to measure nonlinear dependence evolves as the dimensions increase.
The paper also shows that the rescaled HSIC converges in distribution to a standard normal distribution under the null hypothesis and provides the conditions needed to have nontrivial power in high dimensions. The theory is validated by simulations and real-world data involving stock prices from the energy sector and raw material sector in the US stock market.





**Strengths:**

This is a solid paper with strong theory and convincing numerical support. A key advantage of Theorem 1, which provides the asymptotic distribution (with rates) of the HISC test statistic under the null hypothesis, is that it alleviates the use of permutation test to decide critical values, a requirement commonly observed in other tests of independence.

The phase transition of the convergence rates in Theorem 3 is illuminating.


**Weaknesses:**

None.

**Questions:**

How different is the theory between yours and the 2021 paper by Gao et al. in the Annals of Statistics?  What are the differences in technical tools and level of difficulties?

**Limitations:**

Some constraints are imposed in the theory but they are fairly mild and reasonable.

---

> ### Author Rebuttal · Authors · 2023-08-07
>
> We thank you for taking the time to read our paper and for your valuable feedback and constructive suggestions.
>
> Comment 1: How different is the theory between yours and the 2021 paper by Gao et al. in the Annals of Statistics? What are the differences in technical tools and level of difficulties?
>
> Response 1: We appreciate your interest in our paper and the work of Gao et al. (2021). Indeed, our first results are parallel to theirs. In particular, we prove that the rescaled HSIC converges in distribution to a standard normal distribution under the null hypothesis. We also derive a general condition for the HSIC based tests to have power asymptotically approaching one. However, the main difference between our theory and theirs is that we focus on the HSIC, which includes the distance correlation as a special case.
>
> In contrast to their paper, our paper provides a much more extensive analysis of HSIC. We demonstrate that HSIC can detect different kinds of dependences that depend on the dimensionality and sample orders, which is a novel and important insight that has not been explored in the literature before. Furthermore, our proof technique in this part is completely different from theirs, which indicates a higher level of difficulty. This is the main contribution of our paper and sheds light on the performance of HSIC in high dimensions.
>
> References
>
> Gao, L., Fan, Y., Lv, J., and Shao, Q.-M. (2021). Asymptotic distributions of high- dimensional distance correlation inference. The Annals of Statistics, 49(4):1999–2020.

---

### Official Review · Reviewer_Uk8i · 2023-07-04

**Soundness:** 3 good
**Presentation:** 4 excellent
**Contribution:** 3 good
**Rating:** 8
**Confidence:** 3

**Summary:**

The paper provides insights into the properties of HSIC in high dimensions, more specifically the rate at which sample size must grow in order to detect non-linear correlations if data is high dimensional. The results are categorized based on scenarios where either one or both variables have a "growing" dimension, and they express various types of nonlinearity using conditional expected values of higher orders.

A great summary of the results can be found in lines 61 to 64 of the paper.


**Strengths:**

The paper effectively presents the main results and is written with clarity. In my opinion, the results are  significant within the sub-area of independence testing. It has been recognized  for some time that the power of HISC diminishes in high dimensions, but as far as I know, no one has provided conditions on the sample size, dimension and degree of non-linearity for the test to detect signal. Formulation of the results in the language of conditional expected values of higher moments yields insightful and concise characterization of the limitations of HSIC in high dimensions.




**Weaknesses:**

I'm happy to see empirical study (section 5.3) in a theoretical paper. Having said that, I would be surprised if a linear test would not reject the null  (pool the returns per sector and run a correlation based test).

Edit: I read the response, and this still doesn't fully make sense to me. I would have chosen a dataset with a non-linear dependence only  and shown HSIC failing. However, ultimately, this does not detract from the quality of the findings and I stand by 7 (8 makes sense too).

**Questions:**

Is there related work for MMD? Seems like similar results should/could hold for MMD.

**Limitations:**

yes

---

> ### Author Rebuttal · Authors · 2023-08-07
>
> We thank you for taking the time to read our paper and for your valuable feedback and constructive suggestions.
>
> Comment 1: I'm happy to see empirical study (section 5.3) in a theoretical paper. Having said that, I would be surprised if a linear test would not reject the null (pool the returns per sector and run a correlation based test).
>
> Response 1: Indeed, in this example, the monthly mean stock prices of energy companies are linearly dependent with the raw material companies without a doubt. We use this example to confirm the assertions we made in Theorem 3. In particular, according to the second part of Theorem 3 and the discussions at the end of Section 4, the HSIC can only have nontrivial power if $p^{(s_1-1)\kappa_x}q^{(s_2-1)\kappa_y}=o(n)$. Because in this data set, both p and q are much larger than n, this condition is only satisfied if $s_1=s_2=1$, which corresponds to the covariance between x and y. This together with the fact that the test based on HSIC rejected the null hypothesis, we conclude that there exists a linear dependence relationship between x and y. This type of linear dependence is also confirmed by the RV coefficient as well as the $R^2$s we computed for Denison Mines Corp. and Energy Fuels Inc., and Uranium Energy Corp. and Energy Fuels Inc.. We will emphasize the motivations of this analysis more precisely in the revised version. Thank you for your comment. Please let us know if you have any further questions.
>
> Comment 2: Is there related work for MMD? Seems like similar results should/could hold for MMD.
>
> Response 2: Our results focus on the dependence measures between two random vectors, which can be similarly adapted to the two sample test context using MMD. In fact, our results can shed light on the performance of MMD in high dimensions, as we can translate the two sample problems into independence problems. To illustrate this idea intuitively, we define two new random variables U and V as follows: V is either identical to X or Y, and U is set to be 1 if V=X and 0 if V=Y. Then, $f_X(t)=f_Y(t)$ is equivalent to $f_{V\mid U=1}(t) = f_{V\mid U=0}(t)$. This means that U and V are independent, since the conditional distribution of V given U does not depend on the value of U. Therefore, testing whether X and Y have the same distribution is equivalent to testing whether U and V are independent. Note that when X and Y are high-dimensional covariates, V is also high-dimensional and U is univariate. Hence, we can apply the first part of Theorem 3 to understand the behavior of MMD in high dimensions.

---

### Official Review · Reviewer_56Ec · 2023-07-06

**Soundness:** 3 good
**Presentation:** 3 good
**Contribution:** 3 good
**Rating:** 7
**Confidence:** 3

**Summary:**

A paper providing tighter analysis and tests for HSIC statistics for independence in some regimes of interest

NB I have only a nodding acquaintance with this statistic, but have used it and regard it as of high importance. I have done my best to learn the background in the time available.

**Strengths:**

The model provides an interesting and non-trivial bounds for HSIC to test for dependence of high-dimensional covariates, which is an important and useful setting. If I understand correctly they constrain the complexity of polynomial mean dependence that can be detected given covariate sizes and dimensions.

**Weaknesses:**

The conditions are very stringent, and so we are left tow wonder if the results apply to real problems.

the isotropic kernel choice is a very strong restriction, and one that I would never use in practice - heuristically we expect low discrimination power for kernel-based methods under istoropy for "most" kernel methods; it would not be surprsiing if this was true for HSIC in particular. I am not clear how essential the isotropy is but the results start to look like a trivial if that is all they can handle: we might think of these as a "gaussian thin shell"-type result in that case.

Independence is particularly important when it is conditional at which point it gives us Bayesian networks; Do any of these results survive for conditional independence? The authors mention such applications (l34-l37) but I believe thereafter discard them.

I am taking the validity of the proofs here largely on faith. Nothing "looks" odd, but I have not stringently checked.

**Questions:**

Is the isotropy essential to these bounds or can we use other kernels? Do the bound still hold if we relax isotropy to some "better" distance metric? How about if we use a kernel that incorporates some prior knowledge of the domain?
How about other kernels, incl nonstationary ones? Can I improve my bounds by using a polynomial kernel, which is not a characteristic but might be useful for certain types of dependence? Or dot-product-type kernels?

Can we apply these results to conditional independence testing, i.e. inferring graphical models?

If not, the results are still cool, but they do not, IMO, offer anything practical. They would still be a useful improvement in our understanding, however, and thus I favour publishing this.

**Limitations:**

The authors are clear about necessary conditions for their theorems to hold, but sufficient conditions are not obvious to me. See above for questions about generality.

---

> ### Author Rebuttal · Authors · 2023-08-07
>
> We thank you for taking the time to read our paper and for your valuable feedback and constructive suggestions.
>
> Comment 1: Is the isotropy essential to these bounds or can we use other kernels? Do the bound still hold if we relax isotropy to some "better" distance metric? How about if we use a kernel that incorporates some prior knowledge of the domain? How about other kernels, incl nonstationary ones? Can I improve my bounds by using a polynomial kernel, which is not a characteristic but might be useful for certain types of dependence? Or dot-product-type kernels?
>
> Response 1:  Indeed, the isotropy kernel assumption is essential to the bounds derived in this paper. This kind of kernel is very commonly used in the literature and includes many positive-definite kernels such as the Gaussian kernel, the Laplacian kernel, the rational quadratic kernel, and the kernel generating Sobolev spaces, etc. We believe that our results can be similarly generalized to many other kinds of kernels, including non-stationary ones. However, this would require some additional technical assumptions and modifications of our proofs. We leave this as an open problem for future research.
>
> However, there do exist some "better" distance metrics that could potentially improve the performance of our method. We are currently working on this direction, but we cannot share the details publicly at this moment. Please understand. To motivate you further, we can provide some references that explore some of these ideas in different settings. For example, Zhu et al. (2020) suggest to aggregate marginal sample HSIC as the test statistic instead of using HSIC over the whole features. Chakraborty and Zhang (2021) propose a new distance for Euclidean space that is capable of detecting marginal nonlinear dependences in high dimensions.
>
> We hope this answers your question.
>
> Comment 2: Independence is particularly important when it is conditional at which point it gives us Bayesian networks; Do any of these results survive for conditional independence? The authors mention such applications (l34-l37) but I believe thereafter discard them. Can we apply these results to conditional independence testing, i.e. inferring graphical models?
>
> Response 2: Thank you for inspiring us. We believe that the results summarized in this work can be applied to conditional independence testing methods such as the conditional distance correlation (Wang et al., 2015) in high dimensions. However, this would require some extra effort to prove them rigorously. Therefore, we leave this as an open problem for future research.
>
> References
>
> Chakraborty, S. and Zhang, X. (2021). A new framework for distance and kernel-based metrics in high dimensions. Electronic Journal of Statistics, 15(2):5455–5522.
>
> Wang, X., Pan, W., Hu, W., Tian, Y., and Zhang, H. (2015). Conditional distance correlation. Journal of the American Statistical Association, 110(512):1726–1734.
>
> Zhu, C., Zhang, X., Yao, S., and Shao, X. (2020). Distance-based and rkhs-based dependence metrics in high dimension. The Annals of Statistics, 48(6):3366–3394.

---

### Official Review · Reviewer_V7Sb · 2023-07-09

**Soundness:** 2 fair
**Presentation:** 2 fair
**Contribution:** 2 fair
**Rating:** 4
**Confidence:** 3

**Summary:**

The authors provide the statistical properties of HSIC, which is the measure of independency between two random variables. When the random variables have high-dimensionality and have nontrivial dependency, the authors provide the condition for the number of samples in order to successfully detect the dependency.

**Strengths:**

For the two cases when only one variable is high-dimensional and when both variables are high-dimensional, the authors provide the number of data conditions that the nontrivial dependency can be reliably detected. In particular, when there is no lower-order dependency, the authors show that HSIC experience a difficulty to measure the dependence appropriately. Because the analysis is asymptotic, it is not guaranteed that the tendency should appear with finite dimensions. However in the experiments shown in this manuscript, the tendency is clearly shown in both synthetic and real data.

**Weaknesses:**

Choice of kernels should be related to the argument that HSIC only measures linear dependences (l336). What is the effect of the change of \gamma on the equations in Theorem 3 and the experiments?

**Questions:**

1. Is the derived condition for Gaussians (l58-l59)? The authors should mention those conditions in every theorem explicitly.
2. In theorem 2, what is the condition when n*hCorr^2 is finite when n^{1/2}hCorr^2 is infinity when (A1) or (A2) does not hold?
3. What are the curves shown in Figure 1 and Figure 2? What are the horizontal and vertical axes?


**Limitations:**

The comparison with other methods such as distance correlation could provide more information about the difficulty in each case provided in this paper.

---

> ### Author Rebuttal · Authors · 2023-08-07
>
> We thank you for taking the time to read our paper and for your valuable feedback and constructive suggestions.
>
> Comment 1: Choice of kernels should be related to the argument that HSIC only measures linear dependences (l336).
>
> Response 1: Thank you for raising this important point about the choice of kernels. The choice of kernels does not influence the theoretical results provided that the chosen kernels are isotropic and satisfy assumption (A2). It covers many commonly used kernels such as the Gaussian kernel, the Laplacian kernel, the rational quadratic kernel, and the Kernel generating Sobolev spaces, etc. In the empirical study, both dimensions p and q are much larger than n, and this is exactly the case where the conditions of the second part of Theorem 3 hold when $s_1=s_2=1$. In this circumstance, HSIC degenerates to the covariance, which only measures linear dependence. That explains why "HSIC only measures linear dependences" in line 336.
>
> Comment 2: What is the effect of the change of $\gamma$ on the equations in Theorem 3 and the experiments?
>
> Response 2: We appreciate your insightful question about the effect of the change of $\gamma$ on our results. Theoretically, the bandwidth parameter $\gamma$ can be chosen from a wide range of values, as long as it satisfies condition (A2). In practice, we use the median of $||z_1-z_2||$ as a default value for $\gamma$, since it ensures that Assumption (A2) holds for many common kernels. However, our method is robust to different choices of $\gamma$. We will emphasize this in the revised version of the paper. We vary $\gamma$ from $0.5\gamma_m$ to $2\gamma_m$, where $\gamma_m$ is the median of $||z_1-z_2||$, and report the performance of our method on Examples 1 to 3. The results are consistent across different values of $\gamma$, showing the stability and reliability of our approach. For illustration, we present the empirical type-I error rates at the significance level $\alpha=0.05$ for Example 1 with $p=q=10$ in the table below. We hope this answers your question and clarifies our method.
>
> | | 0.5$\gamma_m$ | $\gamma_m$| 1.5$\gamma_m$| 2$\gamma_m$|
> |:------|:------|:------|:------|:------|
> |Gaussian| 0.0548 | 0.0580 | 0.0570 | 0.0554|
> |Laplacian| 0.0530 | 0.0590 | 0.0588 | 0.0586|
>
> Comment 3: Is the derived condition for Gaussians (l58-l59)? The authors should mention those conditions in every theorem explicitly.
>
> Response 3: We would like to clarify that we do not assume that the random vectors are Gaussian in our paper. In lines 58 to 59, the zero mean and identity covariance matrix assumptions are only for illustrative purposes, and they are not essential for our theoretical results. In all the theorems in our paper, we do not impose any assumptions on the distributions of the random vectors. We will clarify this point in the revised version of the paper.
>
> Comment 4: In theorem 2, what is the condition when $n*hCorr_n^2$ is finite when $n^{1/2} hCorr^2$ is infinity when (A1) or (A2) does not hold?
>
> Response 4: Thank you for this question. Theorem 2 shows that if  $n^{1/2}hCorr^2\to\infty$ and Assumptions (A1) and (A2) hold true, the test based on HSIC would have power approaching 1, i.e., for $n * hCorr_n^2$ to diverge to infinity under the alternative hypothesis. This is because $n*hCorr_n^2$ converges to a normal distribution under the null hypothesis.
>
> Assumption (A1) restricts the dependence structures within the coordinates of $z$, while Assumption (A2) imposes some conditions on the kernels and the bandwidth parameters. For example, Assumption (A1) would be violated if all coordinates of $z$ are identical, and Assumption (A2) would be violated if the Gaussian kernel is used and $\gamma_z$ is small enough such that  $E||z_1^*-z_2^*||^2\to\infty$.  We demonstrate these cases below Assumptions (A1) and (A2) in the paper.
>
> As for $n^{1/2} hCorr^2$, it goes to infinity as long as the dependence measured by $hCorr^2$ does not decay to zero too fast. We show in Section 4 that it captures different types of dependences, depending on the dimensionality and sample orders.
>
> Comment 5: What are the curves shown in Figures 1 and Figure 2? What are the horizontal and vertical axes?
>
> Response 5: We apologize for not providing enough details in the captions of Figures 1 and 2. The figures show the kernel densities of the test statistics under the null hypothesis, computed from 5000 simulations. The horizontal axes represent the observed values of the test statistics, and the vertical axes represent the kernel densities of those values. We use two different kernels to implement the tests, namely Gaussian (dashed line) and Laplacian (dotted line). The solid line is the reference curve, which is the density of the standard normal distribution. We will revise the captions to include this information.

---

### Official Review · Reviewer_WrHD · 2023-07-24

**Soundness:** 3 good
**Presentation:** 2 fair
**Contribution:** 2 fair
**Rating:** 6
**Confidence:** 2

**Summary:**

This article deals with the problems of measuring nonlinear dependence between random vectors living in Euclidean spaces and testing for their independence. The authors provide statistical insights into the performance of one of the two major criteria, the Hilbert-Schmidt independence criterion (HSIC), when the dimensions of the random vectors grow at different rates. Their theoretical contribution is completed with an empirical study involving both artificial and real-world data sets.

**Strengths:**

The major strong point of the contribution seems to be the real data application, which could be of interest even to the non specialist.

**Weaknesses:**

The paper will not appear as self-contained to the non specialist (like me).

The naive reader will find it contrary to intuition that the computation of a criterion measuring a basic statistical connection between random vectors should involve the choice of two kernel functions. This is all the more strange as the two vectors take their values in Euclidean spaces, but the Euclidean dot product is not an option to be favoured. Could this be motivated in a simple way?

The originality of the contribution is difficult to assess, all the more since the other major criterion, the distance correlation (DC) criterion, has already been the subject of a similar study.

**Questions:**

Could the authors provide the definition of technical concepts, like the "degree of conditional mean of x given y", or the "n-th Kronecker power of x"? More generally, could they make the paper technically more self-contained for the non specialist?

Could the comparison between HSIC and DC be developed ?

The typos should be corrected.

**Limitations:**

This criterion does not apply here.

---

> ### Author Rebuttal · Authors · 2023-08-07
>
> We thank you for taking the time to read our paper and for your valuable feedback and constructive suggestions.
>
> Comment 1: The paper will not appear as self-contained to the non specialist (like me). The naive reader will find it contrary to intuition that the computation of a criterion measuring a basic statistical connection between random vectors should involve the choice of two kernel functions.
>
> Response 1: We agree that the choice of two kernel functions may seem unintuitive and complicated to some readers, especially those who are not familiar with the HSIC test. However, for the sake of generality and completeness, we allow them to be different in our theoretical study. In practice, one can choose the same kernel function for both variables, or use a common kernel function such as the Gaussian kernel, which has been shown to perform well in many previous studies  (see, e.g., Albert et al., 2022). We hope this clarifies our motivation and rationale for choosing two kernel functions in our paper.
>
> Comment 2: This is all the more strange as the two vectors take their values in Euclidean spaces, but the Euclidean dot product is not an option to be favored. Could this be motivated in a simple way?
>
> Response 2: The Euclidean dot product is not an option to be favored because it does not capture the nonlinear dependence between random variables. For example, the covariance uses the Euclidean dot product, which can only measure linear dependences. The kernel functions, on the other hand, can measure the dependence between the variables in a high-dimensional feature space, where the nonlinear dependence can be better detected. This is the essence of the kernel trick, which is widely used in machine learning and statistics. We hope this explains why we do not use the Euclidean dot product in our paper.
>
> Comment 3: The originality of the contribution is difficult to assess, all the more since the other major criterion, the distance correlation (DC) criterion, has already been the subject of a similar study.
>
> Response 3: We appreciate the reviewer’s comment on the originality of our contribution. We agree that the distance correlation (DC) has been studied in the literature, such as in Zhu et al. (2020) and Gao et al. (2021). However, our paper is different from theirs in the following ways.
> - We study the HSIC, which is more general than DC. We prove its asymptotic normality under the null and a general condition for its consistency under the alternative.
> - We provide a much more comprehensive analysis of the HSIC based test in high dimensions than previous works. We show that HSIC can capture different types of dependences, depending on the dimensionality and sample orders, which have not been realized before. Our results characterize a full picture of the HSIC based test in high dimensions, while previous works only focused on some specific cases.
> We hope this clarifies the originality and contribution of our paper.
>
> Comment 4: Could the authors provide the definition of technical concepts, like the "degree of conditional mean of x given y", or the "n-th Kronecker power of x"? More generally, could they make the paper technically more self-contained for the non specialist?
>
> Response 4: Thank you for the kind reminder. The degree of conditional mean of x given y quantifies the difference between $E(x\mid y)$ and $Ex$, which is measured by $MD(x\mid y)$ in the paper. The n-th Kronecker power of x is defined as  $x^{\otimes n} = x\otimes x^{\otimes(n-1)}$, $x^{\otimes 1} = x$, and $\otimes$ denotes the Kronecker product. We sincerely apologize for any lack of clarity in the previous version of the paper. We will make significant efforts to rephrase technical terms and provide more intuitive explanations to aid non-specialist readers.
>
> Comment 5: Could the comparison between HSIC and DC be developed?
>
> Response 5: We appreciate your interest in comparing HSIC and DC. As we mentioned in the paper, DC is a special case of HSIC when the distance-induced kernel is used, and our results also apply for DC. Therefore, the comparison between HSIC and DC depends on the choice of kernels. However, choosing an appropriate kernel for HSIC is not trivial. Therefore, we believe that there is no definitive answer to which method is better in theory, and the performance may vary depending on the data and the application. We hope this clarifies our point of view.
>
> Comment 6: The typos should be corrected.
>
> Response 6: We appreciate the reviewer's comment on correcting the typos in our manuscript. As suggested, we went through the whole manuscript carefully and made every effort to correct typos and grammatical errors. For instance, we corrected "fiar" to "fair" in line 44 and changed the first parenthesis to braces in line 299.
>
> References
>
> Albert, M., Laurent, B., Marrel, A., and Meynaoui, A. (2022). Adaptive test of inde- pendence based on hsic measures. The Annals of Statistics, 50(2):858–879.
>
> Gao, L., Fan, Y., Lv, J., and Shao, Q.-M. (2021). Asymptotic distributions of high- dimensional distance correlation inference. The Annals of Statistics, 49(4):1999–2020.
>
> Zhu, C., Zhang, X., Yao, S., and Shao, X. (2020). Distance-based and rkhs-based dependence metrics in high dimension. The Annals of Statistics, 48(6):3366–3394.

---

### Decision · Program_Chairs · 2023-09-21

**Decision:**

Accept (poster)

**Comment:**

The submission is centered around measuring and testing statistical independence between two random variables taking values in finite-dimensional Euclidean spaces, using the (normalized) Hilbert-Schmidt independence criterion (HSIC), in the high-dimensional setting. The authors establish the asymptotic normality of HSIC  under the null (Theorem 1) and its non-trivial power (Theorem 2) in case of growing dimension, and provide further insights on the influence of dimension on HSIC (Theorem 3). The theoretical results are accompanied with numerical demonstrations on both synthetic and real-world (financial) benchmarks.

Kernel techniques are at the forefront of machine learning and statistics, with leading role in the estimation of statistical independence measures such as HSIC. The reviewers agreed upon the novelty and the clear interest of the submission for the NeurIPS community, both from theoretical and from empirical perspective.